# NEURAL NETWORKS ON SYMMETRIC SPACES OF NONCOMPACT TYPE

**Xuan Son Nguyen, Shuo Yang, Aymeric Histace**
ETIS, UMR 8051, CY Cergy Paris University, ENSEA, CNRS, France
{xuan-son.nguyen,shuo.yang,aymeric.histace}@ensea.fr

## ABSTRACT

Recent works have demonstrated promising performances of neural networks on hyperbolic spaces and symmetric positive definite (SPD) manifolds. These spaces belong to a family of Riemannian manifolds referred to as symmetric spaces of noncompact type. In this paper, we propose a novel approach for developing neural networks on such spaces. Our approach relies on a unified formulation of the distance from a point to a hyperplane on the considered spaces. We show that some existing formulations of the point-to-hyperplane distance can be recovered by our approach under specific settings. Furthermore, we derive a closed-form expression for the point-to-hyperplane distance in higher-rank symmetric spaces of noncompact type equipped with $G$-invariant Riemannian metrics. The derived distance then serves as a tool to design fully-connected (FC) layers and an attention mechanism for neural networks on the considered spaces. Our approach is validated on challenging benchmarks for image classification, electroencephalogram (EEG) signal classification, image generation, and natural language inference.

## 1 INTRODUCTION

Neural networks in non-Euclidean spaces have become powerful tools for addressing problems in a wide range of domains such as natural language processing (Chami et al., 2019; Ganea et al., 2018b), computer vision (Huang & Gool, 2017; Nguyen et al., 2019), and medicine (Liu et al., 2019). There is a rich existing literature focusing on hyperbolic neural networks (HNNs) due to the ability of hyperbolic spaces to represent hierarchical data with high fidelity in low dimensions (Chami et al., 2021). Other examples of non-Euclidean spaces which have been commonly encountered are SPD manifolds. In this paper, we restrict our attention to neural networks with manifold-valued output.

The concept of hyperplanes has proven useful in the construction of HNNs (Ganea et al., 2018b; Shimizu et al., 2021) and classification algorithms in hyperbolic spaces (Fan et al., 2023). There exist two classes of hyperplanes in hyperbolic spaces, namely, Poincaré hyperplanes (Ganea et al., 2018b; Shimizu et al., 2021) which are identified as sets of geodesics, and horocycles (Fan et al., 2023; Helgason, 1984) which are described as manifolds orthogonal to families of parallel geodesics. Recently, some approaches (Chen et al., 2024a; Nguyen & Yang, 2023; Nguyen et al., 2024) have successfully developed matrix manifold analogs of Poincaré hyperplanes. However, these approaches either work for SPD manifolds associated with special families of Riemannian metrics (Chen et al., 2024a), or require rich algebraic structures of the considered spaces, which limits their generality.

In this paper, we present a novel approach for building neural networks on symmetric spaces of noncompact type (Helgason, 1979). These include hyperbolic spaces and SPD manifolds and are generally regarded as being among the most fundamental and beautiful objects in mathematics (Bridson & Häfliger, 2011; Helgason, 1994). Our contributions are summarized as follows:

- We propose a novel method to construct the point-to-hyperplane distance in symmetric spaces of noncompact type. Compared to Ganea et al. (2018b); Nguyen & Yang (2023); Chen et al. (2024a) which only concern with hyperbolic spaces (Ganea et al., 2018b) or SPD manifolds (Nguyen & Yang, 2023; Chen et al., 2024a), our method deals with all those spaces and gives a unified formulation for this distance.

- We derive an expression for the point-to-hyperplane distance in a symmetric space of non-compact type equipped with a $G$-invariant Riemannian metric.

- We propose FC layers and an attention mechanism for neural networks on symmetric spaces of noncompact type. Within the context of this work, we are the first to develop such building blocks to the best of our knowledge.

- We provide experimental results on image classification, EEG signal classification, image generation, and natural language inference showing the efficacy of our approach.

## 2 RELATED WORKS

### 2.1 HYPERBOLIC SPACES

HNNs have gained growing attention since the seminal work in Ganea et al. (2018b), which was inspired by Lebanon & Lafferty (2004) and proposed hyperbolic analogs of several building blocks of deep neural networks (DNNs). Some missing building blocks in Ganea et al. (2018b) (e.g., FC and convolutional layers) were then introduced in Shimizu et al. (2021). Both the works in Ganea et al. (2018b); Shimizu et al. (2021) rely primarily on the construction of Poincaré hyperplanes. Another concept of hyperplanes on hyperbolic spaces (horocycles) was studied in Fan et al. (2023). This approach derives the distance between a point and a horocycle using horospherical projections, which were originally used for dimensionality reduction in hyperbolic spaces (Chami et al., 2021). Motivated by the impressive performance of graph neural networks (GNNs) (Veličković et al., 2018), GNNs in hyperbolic spaces were also investigated (Chami et al., 2019; Gulcehre et al., 2018).

### 2.2 MATRIX MANIFOLDS

Most existing works concern with neural networks on SPD and Grassmann manifolds, and special orthogonal groups. SPDNet, LieNet, and GrNet were among the first networks designed on those spaces (Huang & Gool, 2017; Huang et al., 2017; 2018). In Brooks et al. (2019); Ju & Guan (2023); Kobler et al. (2022); Nguyen (2021); Nguyen et al. (2019); Pan et al. (2022); Wang et al. (2021), the authors either introduced Riemannian batch normalization layers or improved Bimap layers (Huang & Gool, 2017). The works in López et al. (2021); Nguyen (2022a;b); Nguyen & Yang (2023); Nguyen et al. (2024) leverage rich algebraic structures of SPD and Grassmann manifolds to generalize some basic operations and concepts in Euclidean spaces to these manifolds. Inspired by Nguyen & Yang (2023), the work in Chen et al. (2024a) generalized multinomial logistic regression (MLR) to SPD manifolds under two families of Riemannian metrics.

### 2.3 GENERAL RIEMANNIAN MANIFOLDS

There have also been attempts to develop more general frameworks for Riemannian manifolds. The works in Chakraborty et al. (2020); Zhen et al. (2019) advocated the use of weighted Fréchet mean to build a number of layers (e.g., convolutional and residual layers) for neural networks on Riemannian manifolds. In Katsman et al. (2023), the authors parameterized vector fields to design Riemannian residual neural networks. Our work can be connected to this work as one can use our derived distance to parameterize such vector fields. Extensions of SPD batch normalization layers (Brooks et al., 2019) on Lie groups were also proposed (Chen et al., 2024b).

## 3 MATHEMATICAL BACKGROUND

### 3.1 HYPERBOLIC SPACES AND SPD MANIFOLDS

We briefly discuss the geometries of two families of symmetric spaces commonly encountered in machine learning applications.

**Hyperbolic Spaces** The Poincaré model $\mathbb{B}_m$ of $m$-dimensional hyperbolic geometry is defined by the manifold $\mathbb{B}_m = \{x \in \mathbb{R}^m : \|x\| < 1\}$ equipped with the Riemannian metric $\langle u, v \rangle_x = \frac{4}{(1-\|x\|^2)^2} \langle u, v \rangle$ where $u, v \in \mathbb{R}^m$. The Riemannian distance between two points $x, y \in \mathbb{B}_m$ is given

by $d_{\mathbb{B}}(x, y) = \cosh^{-1}\left(1 + 2\frac{\|x-y\|^2}{(1-\|x\|^2)(1-\|y\|^2)}\right)$. A detailed discussion of hyperbolic spaces from a symmetric space perspective is given in Appendix L.1.

**SPD Manifolds**  Here we consider PEM (Chen et al., 2024c) (see Appendix L.2) which is more general than the well-established Log-Euclidean framework (Arsigny et al., 2005). Let $\mathrm{Sym}_m$ be the space of $m \times m$ symmetric matrices. Under PEM, the SPD manifold $\mathrm{Sym}_m^+$ is defined by $\mathrm{Sym}_m^+ = \{x \in \mathrm{Sym}_m : u^T x u > 0 \text{ for all } u \in \mathbb{R}^m, u \neq \mathbf{0}\}$ equipped with the metric $\langle u, v\rangle_x^\phi = \langle D_x\phi(u), D_x\phi(v)\rangle$, where $\phi : \mathrm{Sym}_m^+ \to \mathrm{Sym}_m$ is a diffeomorphism, $D_x\phi : T_x \mathrm{Sym}_m^+ \to T_{\phi(x)} \mathrm{Sym}_m$ is the directional derivative of map $\phi$ at point $x$, $T_x X$ is the tangent space of $X$ at $x \in X$. The Riemannian distance between two points $x, y \in \mathrm{Sym}_m^+$ is given by $d_{\mathbb{PEM}}(x, y) = \|\phi(x) - \phi(y)\|$. A detailed discussion of SPD manifolds from a symmetric space perspective is given in Appendix L.3.

Existing point-to-hyperplane distances on Riemannian manifolds are generally built for one of the above families of symmetric spaces, except for the composite distance (Helgason, 1984; 1994). However, the use of the composite distance for our purposes is not straightforward since it is a vector-valued distance in higher-rank symmetric spaces (e.g., SPD manifolds). In the following, we develop a unified framework to address this limitation of existing works.

### 3.2 SYMMETRIC SPACES OF NONCOMPACT TYPE

This section briefly recaps important concepts used in the paper. We refer the reader to Ballmann (2012); Bridson & Häfliger (2011); Helgason (1979) for further reading.

Roughly speaking, a symmetric space $X$ is a connected Riemannian manifold which is reflectionally symmetric around any point. That is, for any $x \in X$, there exists a local isometry $s_x$ of $X$ such that $s_x(x) = x$ and the differential $D_x s_x = -\mathrm{id}_{T_x X}$. Every (simply-connected) symmetric space is a Riemannian product of irreducible symmetric spaces. A symmetric space is irreducible, if it cannot be further decomposed into a Riemannian product of symmetric spaces. There are two types of (nonflat) irreducible symmetric spaces: compact type and noncompact type. Those two types are interchanged by Cartan duality. Please refer to Appendix L.4 for further discussion. In the following, we restrict our attention to those of noncompact type.

Formally, let $G$ be a connected noncompact semisimple Lie group with finite center, $K$ be a maximal compact subgroup of $G$. Then the symmetric space of noncompact type $X$ consists of the left cosets

$$X := G/K := \{x = gK | g \in G\}.$$

The action of $G$ on $X = G/K$ is defined as $g[x] = g[hK] = ghK$ for $x = hK \in X$, $g, h \in G$. Let $o$ be the origin $K$ in $X$, then the map $\varphi : gK \mapsto g[o]$ is a diffeomorphism of $G/K$ onto $X$.

Let $G = KAN$ be the Iwasawa decomposition Helgason (1979); Sawyer (2016) of $G$, and let $\mathfrak{g}$ and $\mathfrak{a}$ be the Lie algebras of $G$ and $A$, respectively. For any linear form $\alpha$ on $\mathfrak{a}$, set $\mathfrak{g}_\alpha := \{v \in \mathfrak{g} | \forall u \in \mathfrak{a}, [u, v] = \alpha(u)v\}$. Let $\mathfrak{a}^*$ be the dual space of $\mathfrak{a}$. Then the set of restricted roots is defined by $\Sigma := \{\alpha \in \mathfrak{a}^* \setminus \{0\} | \mathfrak{g}_\alpha \neq \{0\}\}$. The kernel of each restricted root is a hyperplane of $\mathfrak{a}$. A Weyl chamber in $\mathfrak{a}$ is a connected component of $\mathfrak{a} \setminus \cup_{\alpha \in \Sigma} \ker(\alpha)$. We fix a Weyl chamber $\mathfrak{a}^+$ and denote by $\overline{\mathfrak{a}^+}$ its closure.

**Geometric boundary**  In a symmetric space $X$ of noncompact type, boundary (ideal) points can be regarded as generalizations of the concept of directions in Euclidean spaces. Intuitively, boundary points represent directions along which points in $X$ can move toward infinity (Chami et al., 2021). The set of boundary points $\partial X$ of $X$ is referred to as the (geometric) boundary of $X$. For instance, the Poincaré disk model (a model of 2-dimensional hyperbolic geometry) is given by $\mathbb{D} = \{(x_1, x_2) : x_1^2 + x_2^2 < 1\}$ (one can think of this set as the set of all complex numbers with length less than 1, i.e., $\mathbb{D} = \{x \in \mathbb{C} : \|x\| < 1\}$). The boundary $\partial \mathbb{D}$ of $\mathbb{D}$ is the unit circle $\partial \mathbb{D} = \{(x_1, x_2) : x_1^2 + x_2^2 = 1\}$.

Let $d(., .)$ be the distance induced by the Riemannian metric. A geodesic ray in $X$ is a map $\delta : [0, \infty) \to X$ such that $d(\delta(t), \delta(t')) = |t - t'|, \forall t, t' \geq 0$. A geodesic line in $X$ is a map $\delta : \mathbb{R} \to X$ such that $d(\delta(t), \delta(t')) = |t - t'|, \forall t, t' \in \mathbb{R}$. Two geodesic rays $\delta, \delta'$ are said to be asymptotic if

$d(\delta(t), \delta'(t))$ is bounded uniformly in $t$. This is an equivalence relation on the set of geodesic rays in $X$. The set $\partial X$ of boundary points of $X$ is the set of equivalence classes of geodesic rays. The equivalence class of a geodesic ray $\delta$ is denoted by $\delta(\infty)$.

**Angular metric**  For $x \in X$ and $\xi, \xi' \in \partial X$, there exist unique geodesic rays $\delta$ and $\delta'$ which issue from $x$ and lie in the classes $\xi$ and $\xi'$, respectively (Ballmann, 2012). One can then define $\angle_x(\xi, \xi')$ to be the angle at $x$ between $\delta$ and $\delta'$ (see Appendix L.5). The angle $\angle(\xi, \xi')$ is defined as

$$\angle(\xi, \xi') = \sup_{x \in X} \angle_x(\xi, \xi').$$

The function $(\xi, \xi') \mapsto \angle(\xi, \xi')$ defines the angular metric (Bridson & Häfliger, 2011) on $\partial X$.

**Busemann functions**  Busemann functions (coordinates) can be regarded as generalizations of the concept of coordinates in Euclidean spaces. In a Euclidean space, given a point $x$ and a unit vector $w$ (which represents a direction), one has

$$-\langle x, w \rangle = \lim_{t \to \infty} (d(x, tw) - d(0, tw)) = \lim_{t \to \infty} (d(x, tw) - t),$$

where $tw, t > 0$ can be seen as a ray that moves toward infinity in the direction of $w$ as $t \to \infty$. Note that the inner product $\langle x, w \rangle$ gives the coordinate of $x$ in the direction of $w$. This observation can be used to compute coordinates in $X$. Let $\delta : [0, \infty) \to X$ be a (unit-speed) geodesic ray and $\xi = \delta(\infty) \in \partial X$. Then, by replacing $tw$ with geodesic ray $\delta(t)$, one defines the Busemann coordinate of a point $x \in X$ in the direction of $\xi$ as

$$B_\xi(x) = \lim_{t \to \infty} (d(x, \delta(t)) - t).$$

The function $B_\xi : X \to \mathbb{R}$ is called the Busemann function associated to the geodesic ray $\delta$.

**Horocycles**  Like a Euclidean hyperplane which is orthogonal to a family of parallel lines, a horocycle is orthogonal to a family of parallel geodesics (Helgason, 1984; 1994). Thus, horocycles can be regarded as symmetric space analogs of Euclidean hyperplanes. Let $M$ be the centralizer of $A$ in $K$, i.e., $M := C_K(A) := \{k \in K | ka = ak \text{ for all } a \in A\}$. The space $\Xi$ of horocycles can be identified (Helgason, 1994) with

$$\Xi := G/MN := \{\eta = gMN | g \in G\}.$$

**Composite distances**  The notion of composite distance is a symmetric space analog of the Euclidean inner product (Helgason, 1984; 1994). Let $\eta = gMN$ be a horocycle where $g \in G$, and let $g = kan$ where $k \in K$, $a \in A$, and $n \in N$. Then $\xi = kM \in \partial X$ is said to be normal to $\eta$, and $\log(a)$ is the composite distance from the origin $o$ to $\eta$. More generally, if $x = gK \in X$, and $\eta = hMN \in \Xi$ where $g, h \in G$, then $H(g^{-1}h)$ is the composite distance from $x$ to $\eta$, where the map $H : G \to \mathfrak{a}$ is determined by $g_1 = k_1 \exp H(g_1) n_1$ with $g_1 \in G$, $k_1 \in K$, and $n_1 \in N$.

## 4    PROPOSED APPROACH

We define hyperplanes and propose a general formulation for the point-to-hyperplane distance on the considered spaces in Sections 4.1 and 4.2, respectively. We then examine the proposed formulation for hyperbolic spaces and SPD manifolds in Section 4.3. In Section 4.4, our distance is derived for spaces equipped with $G$-invariant Riemannian metrics. In Section 5, we show how to build FC layers and an attention mechanism for neural networks on the considered spaces.

### 4.1    HYPERPLANES ON SYMMETRIC SPACES

In Euclidean space $\mathbb{R}^m$, a hyperplane $\mathcal{H}_{a,b}^E$ is defined by

$$\mathcal{H}_{a,b}^E = \{x \in \mathbb{R}^m : \langle x, a \rangle - b = 0\},$$

where $a \in \mathbb{R}^m \setminus \{\mathbf{0}\}$, $b \in \mathbb{R}$, and $\langle ., . \rangle$ is the Euclidean inner product. The hyperplane $\mathcal{H}_{a,b}^E$ can be reformulated as

$$\mathcal{H}_{a,b}^E = \{x \in \mathbb{R}^m : \langle p - x, a \rangle = 0\},$$

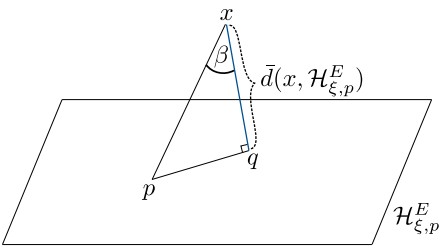

Figure 1: The distance between a point $x$ and a hyperplane $\mathcal{H}^E_{\xi,p}$.

where $p \in \mathbb{R}^m$ and $\langle p, a \rangle = b$.

In order to generalize Euclidean hyperplanes to matrix manifolds, the work in Nguyen & Yang (2023) treats parameter $a$ as a point on the considered manifold $X$. The equation of hyperplane $\mathcal{H}^E_{a,b}$ is then generalized to the matrix manifold setting by defining matrix manifold analogs of operations $-$ and $+$ as well as that of the Euclidean inner product. Here we take a different approach by rewriting $\langle p - x, a \rangle$ as a Busemann function. Let $\xi$ be the equivalence class of the geodesic ray $\delta(t) = t\frac{a}{\|a\|}$, where $\|\cdot\|$ is the Euclidean norm. Using the expression of the Busemann function in $\mathbb{R}^m$ (see Appendix L.6), we have that

$$\langle p - x, \frac{a}{\|a\|}\rangle = B_\xi(-p + x).$$

Assuming that one can define appropriate operations $\ominus$ and $\oplus$ on $X$ that are symmetric space analogs of operations $-$ and $+$, respectively. This leads us to the following definition.

**Definition 4.1 (Hyperplanes on a Symmetric Space).** *For $p \in X$ and $\xi \in \partial X$, hyperplanes on $X$ are defined as*

$$\mathcal{H}_{\xi,p} = \{x \in X : B_\xi(\ominus p \oplus x) = 0\},$$

*where $\ominus$ and $\oplus$ are the inverse and binary operations on $X$, respectively.*

In a symmetric space, a horocycle is a manifold which is orthogonal to families of parallel geodesics (Helgason, 1994). Thus horocycles generalize the idea of Euclidean hyperplanes which are orthogonal to families of parallel lines. In our approach, a hyperplane contains a fixed point $p \in X$ and every point $x \in X$ such that the segment $\ominus p \oplus x$ is orthogonal to a fixed direction $\xi$. Segments of the form $\ominus p \oplus x$ can be regarded as symmetric space analogs of Euclidean lines. Therefore, those hyperplanes also generalize the idea of Euclidean hyperplanes in a natural way.

### 4.2 POINT-TO-HYPERPLANE DISTANCE ON SYMMETRIC SPACES

Let $\mathcal{H}^E_{\xi,p}$ be a hyperplane in $\mathbb{R}^m$. Then the distance $\bar{d}(x, \mathcal{H}^E_{\xi,p})$ between a point $x \in \mathbb{R}^m$ and $\mathcal{H}^E_{\xi,p}$ can be computed (see Fig. 1) as

$$\bar{d}(x, \mathcal{H}^E_{\xi,p}) = d(x, p) \cos(\beta), \tag{1}$$

where $\beta$ is the angle between the segments $[x, p]$ and $[x, q]$ with $q$ being the projection of $x$ on $\mathcal{H}^E_{\xi,p}$. By convention, $\bar{d}(x, \mathcal{H}^E_{\xi,p}) = 0$ for any $x \in \mathcal{H}^E_{\xi,p}$. Note that Eq. (1) can be rewritten as

$$\bar{d}(x, \mathcal{H}^E_{\xi,p}) = d(x, p) \cos \angle_x(\xi', \xi),$$

where $\xi$ and $\xi'$ are the equivalence classes of the geodesic rays $\delta$ and $\delta'$ which issue from $x$ and whose images are the segments $[x, q]$ and $[x, p]$, respectively. Let $x = \delta(t)$, then

$$\bar{d}(x, \mathcal{H}^E_{\xi,p}) = d(x, p) \cos \angle_{\delta(t)}(\xi', \xi) = -d(x, p) \lim_{t \to +\infty} \frac{B_\xi(\delta'(t))}{t}.$$

The last expression (Kapovich et al., 2017) is remarkable because it relates the distance $\bar{d}(.,.)$ to a Busemann function. Note also that

$$B_\xi(\delta'(t)) = -\langle \delta'(t), a \rangle = -\langle ta', a \rangle,$$

where $\delta(t) = ta$, $\delta'(t) = ta'$, $a$ is a unit vector, and $a' = \frac{p-x}{\|p-x\|}$. Therefore

$$\bar{d}(x, \mathcal{H}_{\xi,p}^E) = d(x,p)\langle a', a \rangle = d(x,p)\langle \frac{p-x}{\|p-x\|}, a \rangle = d(x,p)\frac{B_\xi(-p+x)}{\|-p+x\|}.$$

This motivates the following definition.

**Definition 4.2.** *Let* $\mathcal{H}_{\xi,p}$ *be a hyperplane as given in Definition 4.1, and let* $\|\cdot\|_\mathbb{S}$ *be a norm on* $X$. *Then the (signed) distance* $\bar{d}(x, \mathcal{H}_{\xi,p})$ *between a point* $x \in X$ *and* $\mathcal{H}_{\xi,p}$ *is defined as*

$$\bar{d}(x, \mathcal{H}_{\xi,p}) = d(x,p)\frac{B_\xi(\ominus p \oplus x)}{\|\ominus p \oplus x\|_\mathbb{S}}.$$

### 4.3 POINT-TO-HYPERPLANE DISTANCES ON HYPERBOLIC SPACES AND SPD MANIFOLDS

We now derive the point-to-hyperplane distance for the symmetric spaces discussed in Section 3.1.

**Hyperbolic Spaces**   The following result is straightforward.

**Corollary 4.3.** *Let* $\ominus$ *and* $\oplus$ *be the Möbius subtraction* $\ominus_M$ *and Möbius addition* $\oplus_M$ *in* $\mathbb{B}_m$, *respectively, and let* $\|\cdot\|_\mathbb{S}$ *be the Euclidean norm* $\|\cdot\|$ *(see Appendix L.7.1). Let* $p \in \mathbb{B}_m$, $\xi \in \partial\mathbb{B}_m$, *and let* $\mathcal{H}_{\xi,p}$ *be a hyperplane as given in Definition 4.1. Then the distance* $\bar{d}(x, \mathcal{H}_{\xi,p})$ *between a point* $x \in \mathbb{B}_m$ *and* $\mathcal{H}_{\xi,p}$ *is computed by*

$$\bar{d}(x, \mathcal{H}_{\xi,p}) = -\frac{d_\mathbb{B}(x,p)}{\|-p \oplus_M x\|} \log \frac{1 - \|-p \oplus_M x\|^2}{\|-p \oplus_M x - \xi\|^2}.$$

**SPD Manifolds**   Proposition 4.4 shows that the point-to-hyperplane distance studied in Chen et al. (2024a) is a special case of our proposed distance (see Appendix M.1 for the proof of Proposition 4.4).

**Proposition 4.4.** *Let* $\phi : \mathrm{Sym}_m^+ \to \mathrm{Sym}_m$ *be a diffeomorphism. Let* $\oplus$ *and* $\ominus$ *be the binary and inverse operations defined by*

$$x \oplus y = \phi^{-1}(\phi(x) + \phi(y)),$$
$$\ominus x = \phi^{-1}(-\phi(x)),$$

*where* $x, y \in \mathrm{Sym}_m^+$. *Let* $\|\cdot\|_\mathbb{S}$ *be the norm induced by the inner product* $\langle\cdot\rangle_\mathbb{S}$ *given as*

$$\langle x, y\rangle_\mathbb{S} = \langle \phi(x), \phi(y)\rangle.$$

*Let* $\delta(t) = \phi^{-1}(ta)$ *be a geodesic line in* $\mathrm{Sym}_m^+$, *where* $a \in \mathrm{Sym}_m$ *and* $\|a\| = 1$. *Let* $\xi = \delta(\infty)$, $p \in \mathrm{Sym}_m^+$, *and let* $\mathcal{H}_{\xi,p}$ *be a hyperplane as given in Definition 4.1. Then the distance* $\bar{d}(x, \mathcal{H}_{\xi,p})$ *between a point* $x \in \mathrm{Sym}_m^+$ *and* $\mathcal{H}_{\xi,p}$ *is computed as*

$$\bar{d}(x, \mathcal{H}_{\xi,p}) = \langle a, \phi(p) - \phi(x)\rangle.$$

A direct consequence of Proposition 4.4 is that the distance between an SPD matrix and an SPD hypergyroplane (Nguyen & Yang, 2023) is also a special case of our proposed distance under Log-Euclidean and Log-Cholesky frameworks (e.g., the map $\phi$ is the matrix logarithm in the case of Log-Euclidean framework).

### 4.4 POINT-TO-HYPERPLANE DISTANCE ASSOCIATED WITH A $G$-INVARIANT METRIC

In the preceding section, closed-form expressions of the point-to-hyperplane distance are computed for hyperbolic spaces and SPD manifolds under PEM. In this section, we shall derive this distance in a higher-rank symmetric space $X$ of noncompact type equipped with a $G$-invariant Riemannian metric. This requires us (1) to define the binary operation $\oplus$ and inverse operation $\ominus$ on $X$; (2) to define the norm $\|\cdot\|_\mathbb{S}$ on $X$; and (3) to compute the Busemann function.

Let $x = gK$, $y = hK \in X$, where $g, h \in G$.

**Definition 4.5** (**Binary Operation**). *The binary operation* $\oplus$ *is defined as*

$$x \oplus y = ghK.$$

**Definition 4.6** (**Inverse Operation**). *The inverse operation $\ominus$ is defined as*

$$\ominus x = g^{-1}K.$$

The motivation for the above definitions is that the space $G/K$ with the operation $\oplus$ admits a group structure (the identity element is $K$ and the inverse of any element is given by the inverse operation). In order to compute the norm $\| \cdot \|_{\mathbb{S}}$, we shall define an inner product $\langle \cdot, \cdot \rangle_{\mathbb{S}}$ whose construction is based on the following natural view points:

- The inner product $\langle \cdot, \cdot \rangle_{\mathbb{S}}$ should agree with the Riemannian distance.
- The inner product $\langle \cdot, \cdot \rangle_{\mathbb{S}}$ should be invariant under the action of $K$. This property holds for the ones proposed in Helgason (1994); Nguyen & Yang (2023).

We thus consider the following inner product.

**Definition 4.7** (**The Inner Product on Symmetric Spaces**). *Let $x = gK, y = hK \in X$, $g, h \in G$. Then the inner product $\langle \cdot, \cdot \rangle_{\mathbb{S}}$ on $X$ is defined as*

$$\langle x, y \rangle_{\mathbb{S}} = \langle \mu(g), \mu(h) \rangle,$$

*where the map (Cartan projection) $\mu : G \to \overline{\mathfrak{a}^+}$ is determined by $g = k \exp(\mu(g))k'$ with $g \in G$ and $k, k' \in K$ (this follows from the Cartan decomposition Helgason (1979) of $G$ where $\mu(\cdot)$ is a continuous, proper, surjective map to the closed Weyl chamber $\overline{\mathfrak{a}^+}$).*

Proposition 4.8 states that the aforementioned properties hold for the considered inner product (see Appendix M.2 for the proof of Proposition 4.8).

**Proposition 4.8.** *Let $x = gK, y = hK \in X$, $g, h \in G$, and let $\langle \cdot, \cdot \rangle_{\mathbb{S}}$ be the inner product as given in Definitions 4.7. Then*

*(i) We have that:*

$$\| \ominus x \oplus y \|_{\mathbb{S}} = d(x, y),$$

*where the norm $\| \cdot \|_{\mathbb{S}}$ is induced by the inner product $\langle \cdot, \cdot \rangle_{\mathbb{S}}$.*

*(ii) For any $k \in K$, we have that:*

$$\langle x, y \rangle_{\mathbb{S}} = \langle k[x], k[y] \rangle_{\mathbb{S}}.$$

Finally, a closed-form expression of the Busemann function is provided in Proposition 4.9 (see Appendix M.3 for the proof of Proposition 4.9).

**Proposition 4.9.** *Let $\delta(t) = k \exp(ta)K$ be a geodesic ray, where $k \in K$, $a \in \mathfrak{a}$, $\|a\| = 1$, and let $\xi = \delta(\infty)$. Then*

$$B_\xi(x) = \langle a, H(g^{-1}) \rangle,$$

*where $x \in X$, and $g \in G$ is given by $k^{-1}[x] = gK$.*

As a consequence of Proposition 4.9, Corollary 4.10 gives the expression of the distance between a point and a hyperplane in a symmetric space (see Appendix M.4 for the proof of Corollary 4.10).

**Corollary 4.10.** *Let $\delta(t) = k \exp(ta)K$ be a geodesic ray, where $k \in K$, $a \in \mathfrak{a}$, $\|a\| = 1$, and let $\xi = \delta(\infty)$. Let $p = hK \in X$, $h \in G$, and let $\mathcal{H}_{\xi,p}$ be a hyperplane given in Definition 4.1. Then the distance $\bar{d}(x, \mathcal{H}_{\xi,p})$ between a point $x = gK \in X$, $g \in G$ and $\mathcal{H}_{\xi,p}$ is computed as*

$$\bar{d}(x, \mathcal{H}_{\xi,p}) = \langle a, H(g^{-1}hk) \rangle. \tag{2}$$

The connection of the distance in Eq. (2) with existing works is discussed in Appendix G.

## 5    NEURAL NETWORKS ON SYMMETRIC SPACES

In this section, we shall develop symmetric space analogs of two important building blocks in DNNs, i.e., FC layers and attention mechanism. Our starting point is the construction of the point-to-hyperplane distance presented in the preceding section.

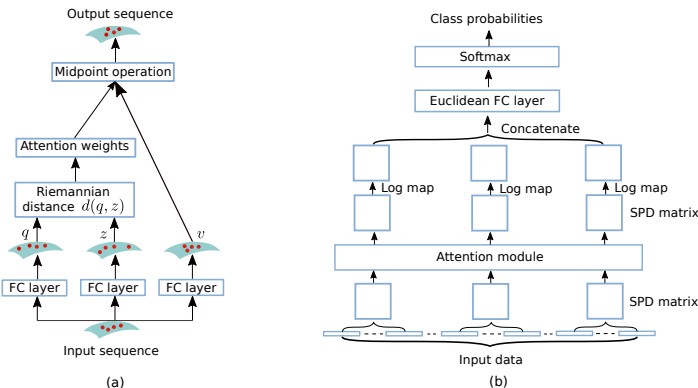

Figure 2: Our proposed attention block (a) and the network architecture for EEG classification (b).

## 5.1 FC LAYERS

An FC layer can be described by the following linear transformation:

$$y = ax - b, \tag{3}$$

where $a \in \mathbb{R}^{m \times m'}$, $x \in \mathbb{R}^{m'}$, and $y, b \in \mathbb{R}^m$. Eq. (3) can be rewritten as a system of equations, each for one dimension, i.e., the $j$-th dimension $y_j, j = 1 \ldots, m$ of the output $y$ is given as

$$y_j = \langle x, a_j \rangle - b_j,$$

where $a_j \in \mathbb{R}^{m'}, b_j \in \mathbb{R}$. Let $\tilde{\xi}_j \in \partial X$ and let $\mathcal{H}_{\tilde{\xi}_j, K}$ be the hyperplane that contains the origin (i.e., $K$) and is orthonormal to the $j$-th axis of the output space. Then $y_j$ can be interpreted as the signed distance $\bar{d}(y, \mathcal{H}_{\tilde{\xi}_j, K})$ from the output $y$ to hyperplane $\mathcal{H}_{\tilde{\xi}_j, K}$. We thus have

$$\bar{d}(y, \mathcal{H}_{\tilde{\xi}_j, K}) = \langle x, a_j \rangle - b_j.$$

From Definition 4.1, we can write the expression $\langle x, a_j \rangle - b_j$ as $B_{\xi_j}(\ominus p_j \oplus x)$, where $p_j \in X$ and $\xi_j \in \partial X$. Therefore

$$\bar{d}(y, \mathcal{H}_{\tilde{\xi}_j, K}) = B_{\xi_j}(\ominus p_j \oplus x). \tag{4}$$

Since the axes of the output space are orthonormal, it is tempting to construct a set of orthonormal boundary points $\{\tilde{\xi}_j\}_{j=1}^m$ for which the output $y$ is related to the input $x$ via Eq. (4). Two boundary points $\tilde{\xi}_l$ and $\tilde{\xi}_j, l, j = 1, \ldots, m, l \neq j$ are said to be orthonormal if $\angle(\tilde{\xi}_l, \tilde{\xi}_j) = \frac{\pi}{2}$. Such a set of boundary points can be identified from Proposition 5.1 (see Appendix M.5 for its proof).

**Proposition 5.1.** *Let $\delta(t) = \exp(ta)K$ and $\delta'(t) = \exp(ta')K$ be geodesic rays, where $a$ and $a'$ are standard basis vectors in $\mathbb{R}^m$, $a \neq a'$. Let $\xi = \delta(\infty)$, $\xi' = \delta'(\infty)$. Then $\xi$ and $\xi'$ are orthonormal.*

We now formulate our proposed FC layers (see Appendix M.6 for the proof of Proposition 5.2).

**Proposition 5.2.** *Let $\delta_j(t) = k_j \exp(ta_j)K, j = 1, \ldots, m$ be geodesic rays, where $k_j \in K$, $a_j \in \mathfrak{a}$, $\|a_j\| = 1$. Let $v_j(x) = B_{\xi_j}(\ominus p_j \oplus x), j = 1, \ldots, m$, where $\xi_j = \delta_j(\infty)$, $p_j \in X$, and $x \in X$ is the input of an FC layer. Then the output $y$ of the FC layer can be expressed as*

$$y = n \exp([-v_1(x) \ldots - v_m(x)])K,$$

*where $n \in N$.*

In our approach, the transformation performed by an FC layer is designed to be a symmetric space analog of the linear transformation in Eq. (3) which makes our approach distinct from existing ones (Huang & Gool, 2017; Huang et al., 2018; Chakraborty et al., 2020; Wang, 2021; Sonoda et al., 2022). Please refer to Appendix H for a comparison of our approach against those approaches.

Table 1: Different formulations of the point-to-hyperplane distance on $\mathbb{B}_m$.

| g-distance | h-distance | b-distance |
|---|---|---|
| $\sinh^{-1}\left(\frac{2\|\langle -p\oplus_M x, a\rangle\|}{(1-\|-p\oplus_M x\|^2)\|a\|}\right)$ | $\frac{1}{a}\left\|a\log\frac{1-\|x\|^2}{\|x-\xi\|^2} - b\right\|$ | $-\frac{d_{\mathbb{B}}(x,p)}{\|-p\oplus_M x\|}\log\frac{1-\|-p\oplus_M x\|^2}{\|-p\oplus_M x-\xi\|^2}$ |
| $x, p \in \mathbb{B}_m, a \in T_p\mathbb{B}_m \setminus \{\mathbf{0}\}$ | $x \in \mathbb{B}_m, a, b \in \mathbb{R}, a > 0, \xi \in \partial\mathbb{B}_m$ | $x, p \in \mathbb{B}_m, \xi \in \partial\mathbb{B}_m$ |
| (Ganea et al., 2018b) | (Fan et al., 2023) | This work |

Table 2: Accuracies (%) of Hybrid ResNet-18 models for image classification.

| Method | CIFAR-10 | CIFAR-100 |
|---|---|---|
| Hybrid Poincaré (Guo et al., 2022) | 95.04±0.13 | 77.19±0.50 |
| Poincaré ResNet (van Spengler et al., 2023) | 94.51±0.15 | 76.60±0.32 |
| Euclidean-Poincaré-H (Fan et al., 2023) | 81.72±7.84 | 44.35±2.93 |
| Euclidean-Poincaré-G (Ganea et al., 2018b) | 95.14±0.11 | **77.78**±0.09 |
| Euclidean-Poincaré-B (Ours) | **95.23**±0.08 | **77.78**±0.15 |

## 5.2 ATTENTION MECHANISM

We use an approach similar to Shimizu et al. (2021). The scaled dot product attention (Vaswani et al., 2017) is formulated as

$$\text{att}(q, z, v) = \text{softmax}\left(\frac{qz^T}{\sqrt{m_z}}\right)v, \qquad (5)$$

where $q, z \in \mathbb{R}^{l \times m_z}$, and $v \in \mathbb{R}^{l \times m_v}$ are the queries, keys, and values, respectively, $l$ is the sequence length, $m_z$ and $m_v$ are the hidden dimensions of the queries (keys) and values, respectively, and function $\text{softmax}(.)$ produces a matrix of the same size as its input matrix by applying the softmax function to each row of this matrix.

The matrix product $qz^T$ corresponds to an attention function that determines the similarities between all query-key pairs. The product of function $\text{softmax}(.)$ and $v = [v_1^T; \ldots; v_l^T]$ produces the weighted means of values $v_j, j = 1, \ldots, l$ and thus can be seen as a midpoint operation. In self-attention, the queries, keys, and values are different linear projections of the same input sequence. Therefore, Eq. (5) can be reformulated as

$$\text{att}(f_{lin}^q, f_{lin}^z, f_{lin}^v, (x_j)_{j=1}^l) = f_{mid}(\{f_{lin}^v(x_j), \pi_{j'j}\}_{j=1}^l)$$

for all $j' = 1, \ldots, l$, where $(x_j)_{j=1}^l$ is the input sequence, $f_{lin}^q(.)$, $f_{lin}^z(.)$, $f_{lin}^v(.)$ are linear functions that project the input points to the queries, keys, and values, respectively, $(\pi_{j'j})_{j=1}^l = \text{softmax}\left(\left(f_{att}(f_{lin}^q(x_{j'}), f_{lin}^z(x_j))\right)_{j=1}^l\right)$, $f_{att}(.,.)$ is the attention function, and $f_{mid}(.)$ is the midpoint operation. We use our proposed FC layers (see Fig. 2 (a)) to perform linear projections in $f_{lin}^q(.)$, $f_{lin}^z(.)$, and $f_{lin}^v(.)$. The attention function (Gulcehre et al., 2018; Shimizu et al., 2021) is given as

$$f_{att}(f_{lin}^q(x_{j'}), f_{lin}^z(x_j)) = -c_1 d(f_{lin}^q(x_{j'}), f_{lin}^z(x_j)) - c_2,$$

where $c_1, c_2 \in \mathbb{R}, c_1 > 0$ are learnable parameters. We adopt the weighted Fréchet mean (wFM) for the midpoint operation.

## 6 EXPERIMENTS

In this section, we report our experimental evaluation on the image classification and EEG signal classification tasks. We refer the reader to Appendix B for experimental details and Appendices C and D for our experimental evaluation on image generation and natural language inference.

Table 3: Accuracies (%) of our networks and state-of-the-art methods for EEG signal classification.

| Method | BCIC-IV-2a | MAMEM-SSVEP-II | BCI-NER |
|---|---|---|---|
| EEG-TCNet (Ingolfsson et al., 2020) | 67.09±4.6 | 55.45±7.6 | 77.05±2.4 |
| MBEEGSE (Altuwaijri et al., 2022) | 64.58±6.0 | 56.45±7.2 | 75.46±2.3 |
| MAtt (Pan et al., 2022) | 74.71±5.0 | 65.50±8.2 | 76.01±2.2 |
| Graph-CSPNet (Ju & Guan, 2023) | 71.95±13.3 | - | - |
| AttSymSpd-LE (Ours) | **78.24** ± 5.4 | **70.96** ± 8.6 | **78.02** ± 2.3 |
| AttSymSpd-GI (Ours) | 78.08 ± 4.8 | 67.24 ± 7.4 | 75.88 ± 2.2 |

## 6.1 HYPERBOLIC SPACES

We follow Bdeir et al. (2024) and design a hybrid architecture[1] which consists of the ResNet-18 (He et al., 2016) and the Poincaré MLR (Ganea et al., 2018b). The output of the ResNet-18 is mapped to the Poincaré ball before it is fed to the Poincaré MLR. We employ our proposed point-to-hyperplane distance as well as those from Ganea et al. (2018b); Fan et al. (2023) (see Tab. 1) in the Poincaré MLR. Experiments are conducted on CIFAR-10 and CIFAR-100 datasets (Krizhevsky, 2009). Tab. 2 shows the results of the three resulting networks and those of Hybrid Poincaré (Guo et al., 2022) and Poincaré ResNet (van Spengler et al., 2023) taken from Bdeir et al. (2024). Hybrid Poincaré only differs from Euclidean-Poincaré-G in the Poincaré MLR which uses the reparameterization method in Shimizu et al. (2021). Our network gives the best mean accuracies on both datasets. In particular, it outperforms all the HNN models from Bdeir et al. (2024) including the fully hyperbolic model on CIFAR-10 dataset. Note that the g-distance is the closest distance from a point to a Poincaré hyperplane, and the b-distance is designed to be a symmetric space analog of the closest distance from a point to a Euclidean hyperplane. However, the h-distance is obtained by horospherical projections which aim to preserve an important property in Principal Component Analysis, i.e., distances between points are invariant to translations along orthogonal directions. Therefore, the h-distance does not have the same nature as the g-distance and b-distance. This probably explains why Euclidean-Poincaré-H is inferior to the other models. The inferior performance of the h-distance can also be observed in our experiments for image generation and natural language inference (see Appendices C and D). Furthermore, those experiments demonstrate that: (1) for image generation, the b-distance outperforms the g-distance in terms of mean performance in all cases; and (2) for natural language inference, the former performs favorably compared to the latter in terms of mean performance in most cases. This indicates that our proposed distance has the potential to improve existing HNNs.

## 6.2 SPD MANIFOLDS

We validate the proposed building blocks (see Section 5) for SPD neural networks on three EEG signal classification datasets: BCIC-IV-2a (Brunner et al., 2008), MAMEM-SSVEP-II (Nikolopoulos, 2021), and BCI-NER (Perrin et al., 2012). We test two variants of the network architecture illustrated in Fig. 2 (b). The FC layers used in the attention block of the first network AttSymSpd-LE are built upon Log-Euclidean metrics (see Section 4.3), while those of the second network AttSymSpd-GI are built upon $G$-invariant metrics (see Sections 4.4 and 5.1).

Tab. 3 shows the results of our networks and some state-of-the-art methods. Most of these methods are selected (Pan et al., 2022) based on two criteria: (1) code availability and completeness; and (2) solid evaluation (e.g., cross-session) without additional auxiliary procedures. As can be observed, AttSymSpd-LE performs the best on all the datasets. AttSymSpd-GI is on par with AttSymSpd-LE on BCIC-IV-2a dataset. Although AttSymSpd-GI is outperformed by AttSymSpd-LE on MAMEM-SSVEP-II and BCI-NER datasets, the former enjoys an advantage of having much smaller numbers of parameters than the latter. For example, AttSymSpd-GI and AttSymSpd-LE use 0.007 MB and 0.034 MB learnable parameters on BCIC-IV-2a dataset, respectively (see also Appendix B.2.3).

---

[1]https://github.com/nguyenxuanson10/symspaces-ic.

ACKNOWLEDGEMENTS

We are grateful for the constructive comments and feedback from the anonymous reviewers.

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

## A  IMPLEMENTATION DETAILS

In this section, we provide details on our implementations of FC layers and the attention module as well as the associated backpropagation procedures.

### A.1  FC LAYERS

*Input:* $x \in X$.

*Trainable parameters:* $k_j \in K$, $a_j \in \mathfrak{a}$, $\|a_j\| = 1$, $p_j \in X$, $j = 1, \ldots, m$, $n \in N$.

*Output:* $y \in X$.

In the case of SPD manifolds, we note that:

- $K = O_m$ (the group of $m \times m$ orthogonal matrices).
- $A$ is the subgroup of $m \times m$ diagonal matrices with positive diagonal entries.
- $N$ is the subgroup of $m \times m$ upper-triangular matrices with diagonal entries 1.

The dimensions of input, output and trainable parameters then can be inferred accordingly ($a_j \in \mathbb{R}^m, j = 1, \ldots, m$).

The computations performed by an FC layer are as follows.

*Step 1:* Compute $\ominus p_j \oplus x = h_j^{-1} g K$ where $p_j = h_j K, x = g K, h_j, g \in G, j = 1, \ldots, m$.

*Step 2:* Compute $g_j \in G, j = 1, \ldots, m$ such that $k_j^{-1}[\ominus p_j \oplus x] = g_j K$.

*Step 3:* Compute $H(g_j^{-1}), j = 1, \ldots, m$ from the Iwasawa decomposition of $g_j^{-1}$, i.e., we need to determine the map $H : G \to \mathfrak{a}$ such that $g_j^{-1} = K \exp(H(g_j^{-1})) N$.

*Step 4:* Compute $B_{\xi_j}(\ominus p_j \oplus x), j = 1, \ldots, m$ as

$$B_{\xi_j}(\ominus p_j \oplus x) = \langle a_j, H(g_j^{-1}) \rangle.$$

*Step 5:* Compute the output of the FC layer as

$$y = n \exp\left(\begin{bmatrix} -B_{\xi_1}(\ominus p_1 \oplus x) & & \cdot \\ & -B_{\xi_2}(\ominus p_2 \oplus x) & \cdot \\ & & -B_{\xi_m}(\ominus p_m \oplus x) \end{bmatrix}\right) K$$

**Backpropagation** Below we desribe the backpropagation procedure for FC layers in the case of SPD manifolds.

*Step 1:* Let $p_j = u_j s_j u_j^T$ and $x = u_x s_x u_x^T$ be eigen decompositions of $p_j$ and $x$, respectively, where $u_j, u_x$ are orthogonal matrices and $s_j, s_x$ are diagonal matrices. Then

$$\ominus p_j \oplus x = s_j^{-\frac{1}{2}} u_j^{-1} u_x s_x^{\frac{1}{2}} K.$$

Eigen decompositions are differentiable operations in the Tensorflow and Pytorch frameworks. Please refer to Ionescu et al. (2015) for a detailed discussion of gradient computations for singular value decompositions and eigen decompositions.

*Step 2:* Compute $g_j = k_j^{-1} s_j^{-\frac{1}{2}} u_j^{-1} u_x s_x^{\frac{1}{2}}$.

*Step 3:* Let $g \in G$ and $g = kan$ with $k \in K$, $a \in A$, and $n \in N$. Then
$$g^T g = n^T a^T k^T kan = n^T a^2 n = n^T a (n^T a)^T.$$

Let $g^T g = cc^T$ be the Cholesky decomposition of $g^T g$ ($c$ is a lower triangular matrix), and let $s$ be the diagonal matrix that contains the main diagonal of $c$. Then $n = (cs^{-1})^T$ and $a = s$. The map $H : G \to \mathfrak{a}$ is then given by $H(g) = \log(a)$. We see that the map $H$ can be determined from a Cholesky decomposition which is a differentiable operation in the Tensorflow and Pytorch frameworks.

*Step 4:* Backpropagation proceeds as normal with the computation of the inner product $\langle a_j, H(g_j^{-1}) \rangle$.

*Step 5:* The output of the FC layer is computed as

$$y = n \exp\left( 2 \begin{bmatrix} -B_{\xi_1}(\ominus p_1 \oplus x) & & \cdot \\ \cdot & -B_{\xi_2}(\ominus p_2 \oplus x) & \cdot \\ \cdot & \cdot & -B_{\xi_m}(\ominus p_m \oplus x) \end{bmatrix} \right) n^T,$$

which involves only a matrix product.

## A.2 ATTENTION MODULE

The pipeline of the attention module is given in Fig. 2(a).

*Input:* A sequence of points $x_j \in X$, $j = 1, \ldots, l$.

*Trainable parameters:* Those include trainable parameters of the three FC layers and $c_1, c_2 \in \mathbb{R}$ (used by the attention function).

*Output:* A sequence of points $y_j \in X$, $j = 1, \ldots, l$.

In the case of SPD manifolds, the dimensions of input, output and trainable parameters follow from those in FC layers (see above).

The computations performed by this module are as follows.

*Step 1:* Apply the three FC layers to the sequence of input points $x_j$ and obtain three sequences of points $(q_j)_{j=1}^l$ (queries), $(z_j)_{j=1}^l$ (keys), and $(v_j)_{j=1}^l$ (values):
$$(q_j)_{j=1}^l = f_{lin}^q((x_j)_{j=1}^l), \quad (z_j)_{j=1}^l = f_{lin}^z((x_j)_{j=1}^l), \quad (v_j)_{j=1}^l = f_{lin}^v((x_j)_{j=1}^l),$$
where $f_{lin}^q(.)$, $f_{lin}^z(.)$, and $f_{lin}^v(.)$ are the linear transformations performed by the three FC layers.

*Step 2:* Compute the similarities between queries and keys $\bar{\pi}_{j'j}, j', j = 1, \ldots, l$ from the Riemannian distance function as
$$\bar{\pi}_{j'j} = -c_1 d(q_{j'}, z_j) - c_2.$$

*Step 3:* Compute the attention weights $\pi_{j'j}, j', j = 1, \ldots, l$ as
$$(\pi_{j'j})_{j=1}^l = \text{softmax}\left((\bar{\pi}_{j'j})_{j=1}^l\right).$$

*Step 4:* Perform the midpoint operation to get the sequence of output points $y_{j'}, j' = 1, \ldots, l$ as
$$y_{j'} = f_{mid}\left(\{v_j, \pi_{j'j}\}_{j=1}^l\right),$$
where $\pi_{j'j}$ is the attention weight associated with $v_j$.

**Backpropagation** We desribe the backpropagation procedure for the attention module in the case of SPD manifolds. Here we only concern with the two blocks "Riemannian distance" and "Midpoint operation". For AttSymSpd-GI, the Riemannian distance used in the "Riemannian distance" block is given by

$$d(x, y) = \| \log(y^{-\frac{1}{2}} x y^{-\frac{1}{2}}) \|,$$

where $x, y \in \mathrm{Sym}_m^+$. The term $y^{-\frac{1}{2}}$ is computed as

$$y^{-\frac{1}{2}} = u s^{-\frac{1}{2}} u^T,$$

where $y = usu^T$ is an eigen decomposition of $y$, $u$ is an orthonormal matrix and $s$ is a diagonal matrix. The $\log(.)$ function is also obtained from an eigen decomposition of its argument.

The "Midpoint operation" block performs the wFM operation under Log-Euclidean framework. Let $\{x_j, w_j\}_{j=1}^L$ be a set of points $x_j \in \mathrm{Sym}_m^+$ with associated weights $w_j \in \mathbb{R}$, where $w_j > 0$ and $\sum_{j=1}^L w_j = 1$. Then the wFM of these points is given by

$$\mathrm{wFM}(\{x_j, w_j\}_{j=1}^L) = \exp\left( \sum_{j=1}^L w_j \log(x_j) \right).$$

The $\exp(.)$ function is computed from an eigen decomposition as $\exp(y) = u \exp(s) u^T$, where $u$ is an orthonormal matrix and $s$ is a diagonal matrix. Thus all functions involved in the computation of wFM are based on eigen decompositions and so backpropagation proceeds as explained above.

# B  EXPERIMENTAL DETAILS

## B.1  IMAGE CLASSIFICATION

### B.1.1  DATASETS

**CIFAR-10 and CIFAR-100 (Krizhevsky, 2009)** CIFAR-10 and CIFAR-100 datasets contain 60K $32 \times 32$ colored images from 10 and 100 different classes, respectively. We use the dataset split implemented in PyTorch, which has 50K training images and 10K testing images.

### B.1.2  EXPERIMENTAL SETTINGS

**Network architecture** Euclidean-Poincaré-G, Euclidean-Poincaré-B, and Euclidean-Poincaré-H have the same architecture which consists of the ResNet-18 and the Poincaré MLR. Here we only present the Poincaré MLR. Let $L$ be the number of classes, then MLR computes the probability of each of the output classes as

$$\mathrm{prop}(y = l | x) = \frac{\exp(a_l^T x - b_l)}{\sum_{j=1}^L \exp(a_j^T x - b_j)} \propto \exp(a_l^T x - b_l), \tag{6}$$

where $x \in \mathbb{R}^m$ is the input, $b_j \in \mathbb{R}$, $a_j \in \mathbb{R}^m, j = 1, \ldots, L$ are model parameters. One can express (Lebanon & Lafferty, 2004) Eq. (6) as

$$\mathrm{prop}(y = l | x) \propto \exp(\mathrm{sign}(a_l^T x - b_l) \|a_l\| \bar{d}(x, \mathcal{H}_{a_l, b_l}^E)), \tag{7}$$

where $\bar{d}(x, \mathcal{H}_{a_l, b_l}^E)$ is the distance between $x$ and hyperplane $\mathcal{H}_{a_l, b_l}^E$ (see Section 4.1). In the Poincaré MLR (Ganea et al., 2018b), Eq (7) is written as

$$\mathrm{prop}(y = l | x) \propto \exp\left( \frac{2}{(1 - \|p_l\|^2)} \|a_l\| \sinh^{-1} \left( \frac{2|\langle -p_l \oplus_M x, a_l \rangle|}{(1 - \| -p_l \oplus_M x\|^2) \|a_l\|} \right) \right), \tag{8}$$

where $x, p_l \in \mathbb{B}_m, a_l \in T_{p_l} \mathbb{B}_m \setminus \{\mathbf{0}\}, l = 1, \ldots, L$.

Euclidean-Poincaré-G uses Eq. (8) to compute the probability of each of the output classes. Euclidean-Poincaré-B and Euclidean-Poincaré-H are constructed by replacing the point-to-hyperplane distance in Eq. (8) with our proposed distance and the one from Fan et al. (2023), respectively.

Table 4: Computation times (seconds) per epoch for image classification experiments (measured on a Quadro RTX 8000 GPU).

| Method | Euclidean-Poincaré-H | Euclidean-Poincaré-G | Euclidean-Poincaré-B |
|---|---|---|---|
| CIFAR-10 | 52 | 57 | 59 |
| CIFAR-100 | 90 | 114 | 118 |

**Hyperparameters**    We follow closely the settings in DeVries & Taylor (2017); Bdeir et al. (2024). Random mirroring and cropping are used for training. The batch size and number of epochs are set to $128$ and $200$, respectively. The learning rate and weight decay are set to $1e - 1$ and $5e - 4$, respectively. The training epochs are set to $60$, $120$, and $160$ for adaptive learning rate scheduling where the gamma factor is set to $0.2$.

**Optimization and evaluation**    All models are implemented in Pytorch. We use the library Geoopt (Kochurov et al., 2020) for Riemannian optimization. RiemannianSGD is used to train the networks. Results are averaged over 5 runs for each model. We use a Quadro RTX 8000 GPU for all experiments.

### B.1.3    MORE RESULTS

The computation times of Euclidean-Poincaré-H, Euclidean-Poincaré-G and Euclidean-Poincaré-B for image classification experiments are presented in Tab. 4. It can be observed that Euclidean-Poincaré-B has high computational costs compared to its competitors, while Euclidean-Poincaré-H is the fastest method among the three methods.

### B.2    EEG SIGNAL CLASSIFICATION

### B.2.1    DATASETS

**BCIC-IV-2a**    It consists of EEG data captured from 9 subjects. The cue-based BCI paradigm consists of 4 different motor imagery tasks, namely the imagination of movement of the left hand (class 1), right hand (class 2), both feet (class 3), and tongue (class 4). Two sessions on different days are recorded for each subject. Each session is comprised of 6 runs separated by short breaks. One run consists of 48 trials (12 trials for each of the 4 possible classes), yielding a total of 288 trials per session. The signals are recorded with 22 Ag/AgCl sensors (with inter-electrode distances of 3.5 cm) and sampled at 250 Hz. They are bandpass-filtered between 0.5 Hz and 100 Hz.

**MAMEM-SSVEP-II**    It consists of EEG data with 256 channels captured from 11 subjects executing a SSVEP-based experimental protocol. Five different frequencies (6.66, 7.50, 8.57, 10.00 and 12.00 Hz) are used for the visual stimulation, and the EGI 300 Geodesic EEG System (GES 300), using a 256-channel HydroCel Geodesic Sensor Net (HCGSN) and a sampling rate of 250 Hz is used to capture the signals.

**BCI-NER**    It consists of EEG data captured from 26 subjects. The EEG electrode placement follows the extended 1020 system. Five sessions (60 trials for the first 4 sessions and 100 trials for the last session) are recorded for each subject, and the duration of a single EEG trial is 1.25 seconds. The signals are recorded with 56 passive Ag/AgCl sensors (VSM-CTF compatible system) and sampled at 600 Hz. Sixteen subjects released in the early stage of the Kaggle competition[2] are used in our experiments.

### B.2.2    EXPERIMENTAL SETTINGS

**Network architecture**    Inspired by Huang & Gool (2017), our network applies a number of convolutional layers to the input data to extract features. The sequence of extracted features is divided into nonoverlapping subsequences, each of them forms an SPD matrix (Huang & Gool, 2017). These

---

[2]https://www.kaggle.com/c/inria-bci-challenge.

procedures create a sequence of SPD matrices, which are fed to the attention block (see Section 5.2). Each output SPD matrix of the attention block is projected to the tangent space at the identity matrix via the logarithmic map (Huang & Gool, 2017). The resulting matrices are transformed into vectors, which are then concatenated to create final features for classification. The network architecture is illustrated in Fig. 2 (b).

For AttSymSpd-LE, we use the distance derived in Proposition 4.4 with $\phi(.) = \log(.)$ to build FC layers in the attention module. As noted in Section 4.3, our definition of hyperplanes and our derived distance match the definition of SPD hypergyroplanes and the distance between an SPD matrix and an SPD hypergyroplane, respectively. Thus we can use the method in Nguyen et al. (2024) for our purposes. Let $x \in \mathrm{Sym}_{m'}^+$ be the input of an FC layer, and let $v_{(l,j)}(x) = \langle \ominus_{le} p_{(l,j)} \oplus_{le} x, a_{(l,j)} \rangle^{le}, p_{(l,j)}, a_{(l,j)} \in \mathrm{Sym}_{m'}^+, l \leq j, l, j = 1, \ldots, m$ (see Appendix L.7.2 for the definitions of operations $\oplus_{le}, \ominus_{le}$ and the SPD inner product $\langle ., . \rangle^{le}$). Then the output $y$ of the FC layer is computed as

$$y = \exp \left( [z_{(l,j)}]_{l,j=1}^m \right),$$

where $z_{(l,j)}$ is given by

$$z_{(l,j)} = \begin{cases} v_{(l,j)}(x), & \text{if } l = j \\ \frac{1}{\sqrt{2}} v_{(l,j)}(x), & \text{if } l < j \\ \frac{1}{\sqrt{2}} v_{(j,l)}(x), & \text{if } l > j \end{cases}$$

For AttSymSpd-GI, the output $y$ of the FC layer in the attention module is computed as

$$y = \exp([-v_1(x) \ldots - v_m(x)])K,$$

where $v_j(x) = B_{\xi_j = \delta_j(\infty)}(\ominus p_j \oplus x), p_j \in \mathrm{Sym}_{m'}^+, \delta_j(t) = \exp(ta_j)K, j = 1, \ldots, m$. For parameters $p_j = g_j K$, we model them on the space of symmetric matrices, and apply the exponential map to obtain SPD matrices (López et al., 2021).

The map $H : G \to \mathfrak{a}$ is computed as follows. Let $g \in G$ and $g = kan$ with $k \in K$, $a \in A$, and $n \in N$. Then $g^T g = n^T a^T k^T kan = n^T a^2 n$, which shows that $a = \exp(H(g))$ and $n$ can be determined from a LDL decomposition of $g^T g$.

To compute the wFM for the midpoint operation, we rely on Log-Euclidean framework. Let $\{x_j, w_j\}_{j=1}^L$ be a set of points $x_j \in \mathrm{Sym}_m^+$ with associated weights $w_j \in \mathbb{R}$, where $w_j > 0$ and $\sum_{j=1}^L w_j = 1$. Then the wFM of these points is given by

$$\mathrm{wFM}(\{x_j, w_j\}_{j=1}^L) = \exp \left( \sum_{j=1}^L w_j \log(x_j) \right).$$

**Hyperparameters** To create sequences of SPD matrices for the attention block, the numbers of nonoverlapping subsequences are set to 4, 6, and 4 on BCIC-IV-2a, MAMEM-SSVEP-II, and BCI-NER datasets, respectively. The number of convolutional layers is set to 2. The numbers of output channels of the first and second convolutional layers are set to 20 and 15, respectively. The sizes of output SPD matrices of FC layers in the attention block are set to $6 \times 6$, $4 \times 4$, and $4 \times 4$ on BCIC-IV-2a, MAMEM-SSVEP-II, and BCI-NER datasets, respectively. The numbers of epochs are set to 400, 100, and 100 on BCIC-IV-2a, MAMEM-SSVEP-II, and BCI-NER datasets, respectively. The batch sizes are set to 128, 64, and 64 on BCIC-IV-2a, MAMEM-SSVEP-II, and BCI-NER datasets, respectively (Pan et al., 2022). The learning rate and weight decay are set to $1e - 3$ and $1e - 1$, respectively.

**Optimization and evaluation** All models are implemented in Tensorflow. Cross-entropy loss and Adam (Kingma & Ba, 2015) are used to train the network. Our evaluation protocol is based on Mane et al. (2021); Pan et al. (2022); Wei et al. (2019). For BCIC-IV-2a dataset, the session 1 data of a subject is used as the training set whose 1/8 is used as the validation set. The session 2 data of the same subject is used as the test set. For MAMEM-SSVEP-II (BCI-NER) dataset, the first 4 sessions of a subject are used as the training set whose 1/4 is used as the validation set. The fifth session of the same subject is used as the test set. In all experiments, the models that obtain the lowest losses on the validation sets are used for testing. The results on BCIC-IV-2a and MAMEM-SSVEP-II datasets are

Table 5: Comparison of the numbers of parameters (MB) of AttSymSpd-GI and AttSymSpd-LE.

| Dataset | BCIC-IV-2a | MAMEM-SSVEP-II | BCI-NER |
|---|---|---|---|
| AttSymSpd-LE | 0.034 | 0.024 | 0.034 |
| AttSymSpd-GI | 0.007 | 0.013 | 0.022 |

Table 6: Effectiveness of the proposed FC layers and attention module on BCIC-IV-2a dataset.

| Method | CovNet | AttSymSpd-GI-Bimap | AttSymSpd-GI |
|---|---|---|---|
| | 73.04±6.34 | 75.82±5.1 | 78.08±4.8 |

computed from accuracies obtained over 10 runs for each subject, while those on BCI-NER dataset are based on the AUC score. Results are averaged over 10 runs for each model. We use a Quadro RTX 8000 GPU for all experiments.

### B.2.3 MORE RESULTS

Tab. 5 reports the numbers of learnable parameters of AttSymSpd-GI and AttSymSpd-LE. Results clearly show that AttSymSpd-GI uses far fewer parameters than AttSymSpd-LE. It is interesting to note that these networks give similar accuracies on BCIC-IV-2a dataset, but AttSymSpd-GI has about $5\times$ fewer parameters than AttSymSpd-LE.

We also study the effectiveness of the proposed FC layers and attention module for EEG signal classification. To this end, we evaluate the performance of AttSymSpd-GI in two cases:

- The attention module is removed from the network. The resulting network is called Cov-Net. This allows us to validate the contribution of the attention module.

- The FC layers in the attention module are replaced with Bimap layers (Huang & Gool, 2017). Bimap layers are referred to as FC convolution-like layers and arguably the most commonly used analogs of FC layers in SPD neural networks. The resulting network is called AttSymSpd-GI-Bimap.

Tab. 6 reports results of our experiments. It can be observed that both the building blocks are effective. In particular, the use of attention module leads to more than 5% improvement in mean accuracy, and the network based on our FC layers outperforms the one based on Bimap layers by more than 2% in terms of mean accuracy.

Finally, Tab. 7 shows all the results from Tab. 3 and additional results of some state-of-the-art methods. It can be seen that AttSymSpd-LE outperforms state-of-the-art methods on all the datasets, while AttSymSpd-GI outperforms them on BCIC-IV-2a and MAMEM-SSVEP-II datasets.

## C IMAGE GENERATION

In this section, we perform image generation experiments using CIFAR-10 and CIFAR-100 datasets. We design a new hyperbolic variational autoencoder (VAE) from HCNN Lorentz (Bdeir et al., 2024) in which we replace the hyperbolic wrapped normal distribution in the Lorentz model with that in the Poincaré ball, and replace the Lorentz MLR with the Poincaré MLR. We use three different point-to-hyperplane distances in the Poincaré MLR as in our image classification experiments.

### C.1 THE LORENTZ MODEL

The Lorentz model $\mathbb{L}_m$ of $m$-dimensional hyperbolic geometry is defined by the manifold $\mathbb{L}_m = \{x = [x_0, \ldots, x_m]^T \in \mathbb{R}^{m+1}, x_0 > 0 : -x_0^2 + \sum_{i=1}^m x_i^2 = -1\}$ equipped with the Riemannian metric $\langle ., . \rangle_x = \mathrm{diag}(-1, \ldots, 1)$. The Riemannian distance between two points $x = [x_0, \ldots, x_m]^T, y = [y_0, \ldots, y_m]^T \in \mathbb{L}_m$ is given by $d_{\mathbb{L}}(x, y) = \cosh^{-1}\left(x_0 y_0 - \sum_{i=1}^m x_i y_i\right)$.

Table 7: Accuracies of our networks and state-of-the-art methods for EEG signal classification.

| Method | BCIC-IV-2a | MAMEM-SSVEP-II | BCI-NER |
|---|---|---|---|
| ShallowNet (Schirrmeister et al., 2017) | 61.84±6.39 | 56.93±6.97 | 71.86±2.64 |
| EEGNet (Lawhern et al., 2018) | 57.43±6.25 | 53.72±7.23 | 74.28±2.47 |
| SCCNet (Wei et al., 2019) | 71.95±5.05 | 62.11±7.70 | 70.93±2.31 |
| EEG-TCNet (Ingolfsson et al., 2020) | 67.09±4.6 | 55.45±7.6 | 77.05±2.4 |
| TCNet-Fusion (Musallam et al., 2021) | 56.52±3.0 | 45.00±6.4 | 70.46±2.9 |
| FBCNet (Mane et al., 2021) | 71.45±4.4 | 53.09±5.6 | 60.47±3.0 |
| MBEEGSE (Altuwaijri et al., 2022) | 64.58±6.0 | 56.45±7.2 | 75.46±2.3 |
| MAtt (Pan et al., 2022) | 74.71±5.0 | 65.50±8.2 | 76.01±2.2 |
| Graph-CSPNet (Ju & Guan, 2023) | 71.95±13.3 | - | - |
| AttSymSpd-LE (Ours) | **78.24** ± 5.4 | **70.96** ± 8.6 | **78.02** ± 2.3 |
| AttSymSpd-GI (Ours) | 78.08 ± 4.8 | 67.24 ± 7.4 | 75.88 ± 2.2 |

Table 8: The network architecture for image generation. The $\text{PROJ}_{\mathbb{R}^m \to \mathbb{L}_m}$ layer maps data in $\mathbb{R}^m$ to $\mathbb{L}_m$. The $\text{H-PROJ}_{\mathbb{L}_m \to \mathbb{B}_m}$ and $\text{H-PROJ}_{\mathbb{B}_m \to \mathbb{L}_m}$ layers map data between $\mathbb{L}_m$ and $\mathbb{B}_m$ (see the text). The CONV and CONVTR layers are Lorentz analogs of the convolutional and transposed convolutional layers, respectively. The BN and RELU layers are Lorentz analogs of the batch normalization and ReLU layers, respectively. The FC-MEAN, FC-VAR, and FC layers are Lorentz analogs of the Euclidean FC layer, respectively. The SAMPLE layer generates random samples in $\mathbb{B}_m$ from the latent distribution of the network. The MLR layer is the Poincaré MLR. Convolutional layers and transposed convolutional layers have kernel sizes of $3 \times 3$ and of $4 \times 4$, respectively. $s$ and $p$ denote stride and zero padding, respectively.

| Layer | CIFAR-10 / CIFAR-100 |
|---|---|
| ENCODER: | |
| $\to \text{PROJ}_{\mathbb{R}^m \to \mathbb{L}_m}$ | $8 \times 8 \times 3$ |
| $\to \text{CONV}_{65,s2,p1} \to \text{BN} \to \text{RELU}$ | $4 \times 4 \times 65$ |
| $\to \text{FLATTEN}$ | 1025 |
| $\to \text{FC-MEAN}_{129}$ | 129 |
| $\to \text{FC-VAR}_{129} \to \text{SOFTPLUS}$ | 129 |
| DECODER: | |
| $\to \text{H-PROJ}_{\mathbb{L}_m \to \mathbb{B}_m}$ | 128 |
| $\to \text{SAMPLE}\ (\mathbb{B}_m)$ | 128 |
| $\to \text{H-PROJ}_{\mathbb{B}_m \to \mathbb{L}_m}$ | 129 |
| $\to \text{FC}_{257} \to \text{BN} \to \text{RELU}$ | 257 |
| $\to \text{RESHAPE}$ | $2 \times 2 \times 65$ |
| $\to \text{CONVTR}_{33,s2,p1} \to \text{BN} \to \text{RELU}$ | $4 \times 4 \times 33$ |
| $\to \text{CONVTR}_{17,s2,p1} \to \text{BN} \to \text{RELU}$ | $8 \times 8 \times 17$ |
| $\to \text{CONV}_{65,s1,p1}$ | $8 \times 8 \times 65$ |
| $\to \text{H-PROJ}_{\mathbb{L}_m \to \mathbb{B}_m}$ | $8 \times 8 \times 64$ |
| $\to \text{MLR}\ (\mathbb{B}_m)$ | $8 \times 8 \times 3$ |

## C.2 NETWORK ARCHITECTURE

The network architecture is given in Tab. 8. The $\text{H-PROJ}_{\mathbb{L}_m \to \mathbb{B}_m}$ layer maps the Lorentz model into the Poincaré ball via the diffeomorphism given as

$$\tau(x_0, x_1, \ldots, x_m) = \frac{(x_1, \ldots, x_m)}{x_0 + 1}.$$

The H-PROJ$_{\mathbb{B}_m \to \mathbb{L}_m}$ layer maps the Poincaré ball into the Lorentz model via the diffeomorphism given as

$$\tau^{-1}(x_1, \ldots, x_m) = \frac{(1 + \|x\|^2, 2x_1, \ldots, 2x_m)}{1 - \|x\|^2}.$$

We briefly present the other layers below. Please refer to Bdeir et al. (2024) for details.

**Lorentz FC layer**

$$y = \text{LFC}(x) = \begin{bmatrix} \sqrt{\|\rho(wx + b)\|^2 + 1} \\ \rho(wx + b) \end{bmatrix},$$

where $x, y$ are the input and output of the layer, respectively, $w \in \mathbb{R}^{m' \times (m+1)}$, and $b \in \mathbb{R}^{m'}$ and $\rho$ denote the bias and activation, respectively.

**Lorentz convolutional layer**   Given an image, the feature of each image pixel is mapped to the Lorentz model. Thus the image can be seen as an ordered set of $m$-dimensional hyperbolic feature vectors. The Lorentz convolution is then performed as

$$y_{h,w} = \text{LFC}(\text{HCat}(\{x_{h'+s\tilde{h}, w'+s\tilde{w}}\}_{\tilde{h}, \tilde{w}=1}^{\tilde{H}, \tilde{W}})),$$

where $\{x_{h'+s\tilde{h}, w'+s\tilde{w}}\}_{\tilde{h}, \tilde{w}=1}^{\tilde{H}, \tilde{W}}$ are the features within the receptive field of the kernel, $\text{HCat}(.)$ denotes the concatenation of hyperbolic vectors, $(h', w')$ denotes the starting position, and $s$ is the stride parameter.

**Lorentz transposed convolutional layer**   The transposed convolutional layer works by swapping the forward and backward passes of the convolutional layer. This is achieved in the Lorentz model through origin padding between the features.

**Lorentz batch normalization**   Given a batch $\mathcal{B}$ of $m$ features $x_i$, the traditional batch normalization algorithm can be described as

$$\text{BN}(x_i) = u \odot \frac{x_i - \text{mean}(\mathcal{B})}{\sqrt{\text{var}(\mathcal{B}) + \epsilon}} + v,$$

where $\text{mean}(\mathcal{B}) = \frac{1}{m} \sum_{i=1}^{m} x_i$, $\text{var}(\mathcal{B}) = \frac{1}{m} \sum_{i=1}^{m} (x_i - \text{mean}(\mathcal{B}))^2$, $u$ and $v$ are parameters to re-scale and re-center the features.

For the Lorentz batch normalization layer, the Lorentzian centroid and the parallel transport operation are used for re-centering, and the Fréchet variance and straight geodesics at the origin's tangent space are used for re-scaling.

**Lorentz ReLU**

$$y = \begin{bmatrix} \sqrt{\|\text{ReLU}([x_1, \ldots, x_m])\|^2 + 1} \\ \text{ReLU}([x_1, \ldots, x_m]) \end{bmatrix},$$

where $x = [x_0, \ldots, x_m]$ and $y$ are the input and output of the layer, respectively.

**Wrapped normal distribution**   The SAMPLE layer uses the method in Nagano et al. (2019) to generate random samples on $\mathbb{B}_m$. Given a normal distribution parameterized by a hyperbolic mean vector $h \in \mathbb{B}_m$ and a Euclidean variance matrix $\Sigma \in \mathbb{R}^{m \times m}$, the layer performs the following operations:

1. Sample a Euclidean vector $\tilde{v}$ from the normal distribution $\mathcal{N}(0, \Sigma)$.
2. Compute $v = \frac{\tilde{v}}{2}$.
3. Parallel transport $v$ from the tangent space of the origin $\mathbf{0}$ to the tangent space of the hyperbolic mean $h$ to obtain a tangent vector $u \in T_h \mathbb{B}_m$ as

$$u = \mathcal{T}_{\mathbf{0} \to h}(v) = (1 - \|h\|^2)v.$$

Table 9: Reconstruction and generation FID of hyperbolic VAEs (lower is better).

| Method | CIFAR-10 | | CIFAR-100 | |
|---|---|---|---|---|
| | Rec. FID | Gen. FID | Rec. FID | Gen. FID |
| Lorentz-Poincaré-H (Fan et al., 2023) | 125.53±5.94 | 69.11±1.67 | 110.36±11.50 | 62.32±6.34 |
| Lorentz-Poincaré-G (Ganea et al., 2018b) | 39.68±1.45 | 49.91±2.06 | 42.82±2.48 | 60.24±4.01 |
| Lorentz-Poincaré-B (Ours) | **38.32**±2.11 | **48.45**±1.31 | **42.05**±2.58 | **59.76**±1.81 |

4. Map $u$ to $\mathbb{B}_m$ by applying the exponential map as

$$z = \exp_h(u) = h \oplus_M \left( \tanh \left( \frac{\|u\|}{1 - \|h\|^2} \right) \frac{u}{\|u\|} \right),$$

where $\oplus_M$ is the Möbius addition (see Appendix L.7.1), and $z$ is the final sample in $\mathbb{B}_m$.

## C.3 EXPERIMENTAL SETTINGS

**Hyperparameters** We adopt the hyperparameters from Bdeir et al. (2024). The curvature for the Lorentz model and the Poincaré ball is set to $1$. The learning rate and weight decay are set to $5e-4$ and $0$, respectively. The batch size and number of epochs are set to $100$. The KL loss weight is set to $0.024$.

**Optimization and evaluation** All models are implemented in Pytorch. We use the library Geoopt (Kochurov et al., 2020) for Riemannian optimization. RiemannianAdam is used to train the networks. We use the reconstruction FID and generation FID to evaluate the networks. The reconstruction FID is computed by comparing test images with reconstructed validation images. A fixed random portion of $10K$ images in the training set is used as the validation set (Bdeir et al., 2024). The generation FID is computed by generating random images from the latent distribution and comparing them with the test set. Results are averaged over 5 runs for each model. We use a Quadro RTX 8000 GPU for all experiments.

## C.4 RESULTS

Results are shown in Tab. 9. Our method achieves the best performances in terms of mean reconstruction FID and mean generation FID in all cases. We can also observe that the h-distance is significantly outperformed by the g-distance and b-distance in these experiments.

## D NATURAL LANGUAGE INFERENCE

In this section, we compare our method for constructing the point-to-hyperplane distance in a Poincaré ball against those in Ganea et al. (2018b); Fan et al. (2023) by performing the same experiments in Ganea et al. (2018b) for textual entailment and detection of noisy prefixes. For the first task, one has to predict whether a sentence can be inferred from another sentence. The second task consists of determining if a sentence is a noisy prefix of another sentence. Experiments are conducted on SNLI (Bowman et al., 2015) and PREFIX datasets (Ganea et al., 2018b) for the first and second tasks, respectively. Our implementation[3] is based on the open-source implementation[4] of HypGRU (Ganea et al., 2018b) that uses the Poincaré MLR as a classification layer. The competing networks differ only in the computation of the point-to-hyperplane distance (see Tab. 1) in the Poincaré MLR.

---

[3] https://github.com/sohata24/nli.
[4] https://github.com/dalab/hyperbolic_nn.

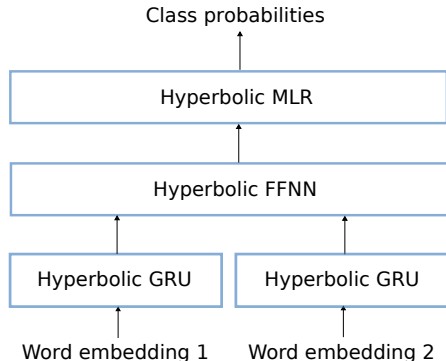

Figure 3: The network architecture for natural language inference.

### D.1 DATASETS

**SNLI (Bowman et al., 2015)**   It consists of 570K training, 10K validation and 10K test sentence pairs. Similarly to Ganea et al. (2018b), The "contradiction" and "neutral" classes are merged into a single class of negative sentence pairs, while the "entailment" class gives the positive pairs.

**PREFIX (Ganea et al., 2018b)**   PREFIX-10%, PREFIX-30%, and PREFIX-50% are synthetic datasets, each of them consists of 500K training, 10K validation, and 10K test pairs. Each dataset is built as follows. For each random first sentence of random length at most 20 and one random prefix of it, a second positive sentence is generated by randomly replacing Z% (Z is 10, 30, or 50) of the words of the prefix, and a second negative sentence of same length is randomly generated. Word vocabulary size is 100.

### D.2 EXPERIMENTAL SETTINGS

**Network architecture**   We use the architecture of the fully hyperbolic GRU (Ganea et al., 2018b) illustrated in Fig. 3. The network consists of a hyperbolic GRU, a hyperbolic feed forward neural network (FFNN), and the Poincaré MLR (see Appendix B.1.2). The update equations of the hyperbolic GRU are given as

$$r_t = \sigma\big(\log_{\mathbf{0}}(W^r \otimes_M h_{t-1} \oplus_M U^r \otimes_M x_t \oplus_M b^r)\big),$$

$$z_t = \sigma\big(\log_{\mathbf{0}}(W^z \otimes_M h_{t-1} \oplus_M U^z \otimes_M x_t \oplus_M b^z)\big),$$

$$\tilde{h}_t = \psi^{\otimes}((W \operatorname{diag}(r_t)) \otimes_M h_{t-1} \oplus_M U \otimes_M x_t \oplus_M b),$$

$$h_t = h_{t-1} \oplus_M \operatorname{diag}(z_t) \otimes_M (-h_{t-1} \oplus_M \tilde{h}_t).$$

where $x_t \in \mathbb{B}_{m_1}$ is the input at frame $t$, $h_{t-1}, h_t \in \mathbb{B}_{m_2}$ are the hidden states at frames $t-1$ and $t$, respectively, $W^r, W^z, W \in \mathbb{R}^{m_2 \times m_2}$, $b^r, b^z, b \in \mathbb{B}_{m_2}$, $U^r, U^z, U \in \mathbb{R}^{m_1 \times m_2}$ are model parameters, $\oplus_M$ is the Möbius addition (see Appendix L.7.1), $\otimes_M$ is the Möbius matrix-vector multiplication (see Appendix L.7.1), $\sigma, \psi$ are activation functions, $\log_{\mathbf{0}}(.)$ is the logarithmic map at $\mathbf{0}$ (see Appendix L.7.1), and the function $\psi^{\otimes}(.)$ is defined as

$$\psi^{\otimes}(x) = \exp_{\mathbf{0}}(\psi(\log_{\mathbf{0}}(x))),$$

where $x \in \mathbb{B}_m$ and $\exp_{\mathbf{0}}(.)$ is the exponential map at $\mathbf{0}$ (see Appendix L.7.1).

Let $x_1, x_2 \in \mathbb{B}_{m_2}$ be the outputs of the hyperbolic GRU corresponding to the first and second sentences, respectively. Then the output $z$ of the hyperbolic FFNN is computed as

$$z = \tau(W_1 \otimes_M x_1 \oplus_M W_2 \otimes_M x_2 \oplus_M b_1 \oplus_M d_{\mathbb{B}}(x_1, x_2) \otimes_M b_2),$$

where $W_1, W_2 \in \mathbb{R}^{m_2 \times m_3}$, $b_1, b_2 \in \mathbb{B}_{m_3}$ are model parameters, $\otimes_M$ is the Möbius scalar multiplication (see Appendix L.7.1), and $\tau$ is an activation function.

Table 10: Accuracies of the competing networks for natural language inference.

| Method | SNLI | PREFIX-10% | PREFIX-30% | PREFIX-50% |
|---|---|---|---|---|
| HypGRU (Ganea et al., 2018b) | 80.89±0.17 | 96.75±0.40 | 87.59±0.46 | **76.45**±0.61 |
| HypGRU-H (Fan et al., 2023) | 80.66±0.46 | 92.20±8.33 | 83.29±6.34 | 74.67±3.12 |
| HypGRU-B (Ours) | **81.01**±0.35 | **97.03**±0.11 | **87.69**±0.04 | 76.25±0.07 |

**Hyperparameters** The word and hidden state embedding dimensions as well as the number of output channels of the hyperbolic FFNN are set to $5$. The number of epochs and batch size are set to $30$ and $64$, respectively. The learning rates for word embeddings and hyperbolic weights are set to $1e-1$ and $1e-2$, respectively. All activation functions in the hyperbolic GRU and hyperbolic FFNN are set to the identity function.

**Optimization and evaluation** We use cross-entropy as the loss function. Euclidean and hyperbolic parameters are optimized using Adam (the learning rate is set to $1e-3$) and Riemannian stochastic gradient descent (RSGD) (Bonnabel, 2013; Ganea et al., 2018a), respectively. Results are averaged over $5$ runs for each model. We use a Quadro RTX 8000 GPU for all experiments.

### D.3 RESULTS

Results of the competing networks are shown in Tab. 10. Our network outperforms HypGRU-H in terms of mean accuracy and standard deviation on all the datasets. Results also demonstrate that our proposed distance has the potential to improve existing HNNs on the considered task.

## E NODE CLASSIFICATION

In this section, we compare our method for constructing the point-to-hyperplane distance in a Poincaré ball against those in Ganea et al. (2018b); Fan et al. (2023) by performing node classification experiments.

### E.1 DATASETS

**Disease (Chami et al., 2019)** It is the transductive variant of a dataset created by simulating the SIR disease spreading model where the label of a node is whether the node was infected or not, and node features indicate the susceptibility to the disease.

**Airport (Chami et al., 2019)** It is a flight network dataset from OpenFlights.org where nodes represent airports, edges represent the airline Routes, and node labels are the populations of the country where the airport belongs.

**Pubmed Namata et al. (2012)** It is a standard benchmark describing citation networks where nodes represent scientific papers in the area of medicine, edges are citations between them, and node labels are academic (sub)areas.

**Cora Sen et al. (2008)** It is a citation network where nodes represent scientific papers in the area of machine learning, edges are citations between them, and node labels are academic (sub)areas. Each publication in the dataset is described by a 0/1-valued word vector indicating the absence/presence of the corresponding word from the dictionary.

The statistics of the four datasets are summarized in Tab. 11. Following Chami et al. (2019), we use 30/10/60 percent splits for Disease dataset, 70/15/15 percent splits for Airport dataset, and standard splits (Kipf & Welling, 2017) with 20 train examples per class for Pubmed and Cora datasets.

| Dataset | #Nodes | #Edges | #Classes | #Features |
|---------|--------|--------|----------|-----------|
| Disease | 1044 | 1043 | 2 | 1000 |
| Airport | 3188 | 18631 | 4 | 4 |
| Pubmed | 19717 | 44338 | 3 | 500 |
| Cora | 2708 | 5429 | 7 | 1433 |

Table 11: Description of the datasets for node classification experiments.

| Dataset Hyperbolicity $\delta$ | Disease $\delta = 0$ | Airport $\delta = 1$ | Pubmed $\delta = 3.5$ | Cora $\delta = 11$ |
|---------|---------|---------|---------|---------|
| HGCN-H | $85.67 \pm 2.58$ | $69.82 \pm 2.08$ | $75.26 \pm 1.82$ | $77.09 \pm 2.02$ |
| HGCN-G | $88.98 \pm 1.96$ | $84.78 \pm 1.48$ | $\mathbf{76.02 \pm 1.09}$ | $77.47 \pm 1.15$ |
| HGCN-B (Ours) | $\mathbf{89.05 \pm 0.78}$ | $\mathbf{85.04 \pm 0.97}$ | $75.89 \pm 0.78$ | $\mathbf{77.90 \pm 1.00}$ |

Table 12: Results of HGCN models for node classification. HGCN-H, HGCN-G, and HGCN-B are built on h-distance, g-distance, and b-distance, respectively. A lower hyperbolicity value $\delta$ means more hyperbolic.

### E.2    EXPERIMENTAL SETTINGS

**Network architecture**    We use the HGCN architecture[5] in Chami et al. (2019) which consists of a graph convolutional network (GCN) and a MLR as the final layer for classification. Both the GCN and the MLR layer are built on the Poincaré ball.

**Hyperparameters**    We follow closely the experimental settings in Chami et al. (2019). The number of epochs is set to 5000. We use early stopping based on validation set performance with a patience of 100 epochs. The learning rate is set to $1e - 2$ for all experiments. The weight decays for Diease and Airport datasets are set to 0. The weight decays for Pubmed and Cora datasets are set to $1e - 4$ and $1e - 3$, respectively. The number of dimensions is set to 16. The number of layers in the GCN is set to 3.

**Optimization and evaluation**    The network is implemented in Pytorch and is trained using cross-entropy loss and Adam optimizer. Results are averaged over 10 random parameter initializations on the final test set.

### E.3    RESULTS

Table 12 shows results of our experiments. We can observe that the b-distance gives the best performance in most cases, which is similar to our results on natural language inference experiments. It can also be observed that the performance of h-distance can nearly match those of g-distance and h-distance on less hyperbolic datasets (high hyperbolicity values), i.e., Pubmed and Cora datasets. However, the h-distance is significantly outperformed by the g-distance and b-distance on more hyperbolic datasets, i.e., Disease and Airport datasets. For those datasets, the b-distance achieves the best performances, demonstrating its effectiveness for datasets with stronger hierarchical structures. Furthermore, in all cases, the b-distance achieve the lowest standard deviation, suggesting that it can offer stable results in the considered application.

## F    COMPLEXITY ANALYSIS

We analyze the complexities of the two network building blocks based on $G$-invariant metrics on SPD manifolds. Let $d_{in}$ and $d_{out}$ be the dimensions of input and output matrices of an FC layer, respectively. Let $l$ be the number of input matrices of the attention module.

---

[5]https://github.com/nguyenxuanson10/symspaces-nc.

- FC layer: It has memory complexity $O(d_{in}^2 d_{out})$ and time complexity $O(d_{in}^3 d_{out})$.
- Attention module: It has memory complexity $O(3d_{in}^2 d_{out})$. The time complexity of the FC layers is $O(3ld_{in}^3 d_{out})$. Both the "Riemannian distance" and "Midpoint operation" blocks have time complexity $O(l^2 d_{out}^3)$.

# G  CONNECTION OF OUR DERIVED POINT-TO-HYPERPLANE DISTANCE WITH EXISTING WORKS

In the case where $\delta(t) = \exp(ta)K$, the distance given in Eq. (2) has a direct connection with the composite distance (see Section 3.2). That is, the former is obtained by the action of functional $a \in \mathfrak{a}$ on the latter, which is a vector-valued distance. This is similar to how the Helgason-Fourier transform (Helgason, 1994) of a function on $X$ is formed. In particular, for any function $f$ on $X$, its Helgason-Fourier transform is defined as

$$\tilde{f}(\lambda, \xi) = \int_X f(x) \exp((-i\lambda + \varrho)A(x, \xi))dx,$$

where $\lambda, \varrho \in \mathfrak{a}^*$, $\xi \in \partial X$, and $A(x, \xi) \in \mathfrak{a}$ denotes the composite distance from the origin $o$ to the horocycle passing through the point $x \in X$ with normal $\xi$. Here the exponent $(-i\lambda + \varrho)\langle x, \xi \rangle_{\mathbb{H}}$ is regarded as the action of functional $-i\lambda + \varrho \in \mathfrak{a}^*$ on the vector-valued distance $A(x, \xi)$ (Helgason, 1994; Sonoda et al., 2022).

The distance given in Eq. (2) is also closely related to random features on hyperbolic spaces (Yu & Sa, 2023). Those features are generated from a map $\mathrm{HyLa}(.) : \mathbb{B}_m \to \mathbb{R}$ given as

$$\mathrm{HyLa}_{\lambda, b, \xi}(x) = \exp\left(\frac{m-1}{2}A(x, \xi)\right)\cos(\lambda A(x, \xi) + b),$$

where $x \in \mathbb{B}_m$, $\xi \in \partial\mathbb{B}_m$, and $\lambda, b \in \mathbb{R}$. By generating $m'$ random samples of tuple $(\lambda, b, \xi)$ from appropriate distributions, one obtains a feature map $\omega : \mathbb{B}_m \to \mathbb{R}^{m'}$ that approximates an isometry-invariant kernel over hyperbolic space $\mathbb{B}_m$. In the higher-rank symmetric space setting, the term $\lambda A(x, \xi)$ will be replaced[6] with the Euclidean inner product of a vector and a vector-valued distance, resulting in a similar formulation of the distance in Eq. (2).

# H  COMPARISON OF OUR FC LAYERS AGAINST EXISTING ONES

In Huang & Gool (2017); Huang et al. (2018), the authors introduce Bimap and FRMap layers and refer them as FC convolution-like layers. For both types of layers, each element of the output matrix is a linear combination of the elements of the input matrix, which is not the case in our FC layers. FRMap layers and our FC layers also differ in their outputs as the former do not produce points on the considered manifolds.

In Chakraborty et al. (2020), the weighted Fréchet Mean (wFM) is adopted to develop Riemannian convolutional layers. These layers cannot be easily extended to build natural generalizations of Euclidean FC layers by simply treating FC layers as special cases of convolutional layers with full kernel size. This is because the resulting FC layers will take as input a set of points on the considered manifold and therefore have no obvious connection with the linear transformation in Eq. (3).

FC layers for neural networks on hyperbolic spaces (Wang, 2021) and symmetric spaces (Sonoda et al., 2022) include activation functions which are not used in our FC layers. Also, FC layers in Sonoda et al. (2022); Wang (2021) do not output points on the considered spaces.

In Nguyen et al. (2024); Shimizu et al. (2021), FC layers are not built upon Busemann functions. Although the method in Nguyen et al. (2024) is designed for matrix manifolds, some of which are not covered by our method (e.g., Grassmann manifolds), the former relies on differentiable forms of certain geometric quantities (e.g., the logarithmic map and parallel transport) which are not required by the latter.

Another line of work (Cohen et al., 2018; Weiler et al., 2021) develops Riemannian neural networks which are functions $f : X \to \mathbb{R}^m$, where $m$ is the number of output channels. The context of these works is different from ours since we aim to build neural networks which are functions $f : X \to X$.

---

[6]Other adaptations are also needed but they are beyond the scope of our paper.

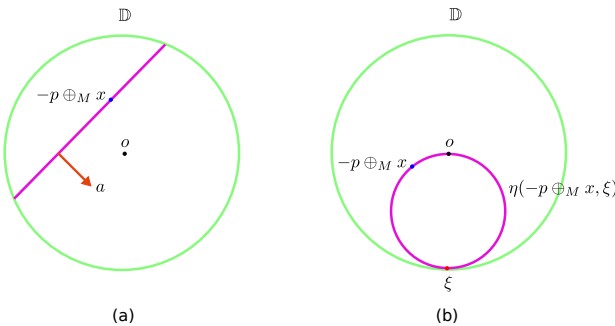

Figure 4: Comparison of Poincaré hyperplanes and our hyperplanes in the Poincaré disk model.
.

## I  ADVANTAGES AND LIMITATIONS OF $G$-INVARIANT AND PEM METRICS

Our motivation for using G-invariant metrics is that such metrics always exist on the considered spaces (e.g., those induced from the Killing form (Helgason, 1979)). As a result, the framework proposed in Section 4.4 is valid for all those spaces. Since Busemann functions associated with $G$-invariant metrics are determined (see the proof of Proposition 4.9) by Busemann functions on the maximal abelian $A$ of $G$ (recall that $G = KAN$) which has much lower dimension than $X$, a major advantage of $G$-invariant metrics compared to PEM metrics is that the former allow FC layers to scale better to high-dimensional input and output matrices. However, due to the same reason, Busemann functions associated with $G$-invariant metrics do not fully capture the structure of $G$ which might lead to poor performance.

The use of PEM metrics is motivated by the fact that like Log-Euclidean metrics, they turn SPD manifolds into flat spaces (i.e., their sectional curvature is null everywhere) which considerably simplifies computation and analysis (e.g., Riemannian computations become Euclidean computations in the logarithmic domain (Arsigny et al., 2005)). PEM metrics are also more general than Log-Euclidean ones. On the downside, those metrics do not yield full affine-invariance (Arsigny et al., 2005; Lin, 2019). This might break the geometric stability principle that plays a crucial role in geometric deep learning architectures (Bronstein et al., 2017).

## J  COMPARISON OF THE B-DISTANCE AND G-DISTANCE

The difference between the b-distance and g-distance can be best explained in the Poincaré disk model $\mathbb{D}$ (see Appendix L.1). A Poincaré hyperplane (Ganea et al., 2018b) is defined as

$$\mathcal{H}_{a,p} = \{x \in \mathbb{D} : \langle \log_p(x), a \rangle_p = 0\},$$

where $p \in \mathbb{D}$, $a \in T_p\mathbb{D} \setminus \{\mathbf{0}\}$, and $\langle .,. \rangle_p$ is the Riemannian metric of the Poincaré disk model. It has been shown (Ganea et al., 2018b) that hyperplane $\mathcal{H}_{a,p}$ can also be described as

$$\mathcal{H}_{a,p} = \{x \in \mathbb{D} : \langle -p \oplus_M x, a \rangle = 0\},$$

which is illustrated by the segment (in purple) in Fig. 4(a).

Since we use the Möbius addition $\oplus_M$ to define the binary operation $\oplus$, a hyperplane in our approach is characterized by

$$\mathcal{H}_{\xi,p} = \{x \in \mathbb{D} : B_\xi(-p \oplus_M x) = 0\},$$

where $p \in \mathbb{D}$ and $\xi \in \partial\mathbb{D}$. One can interpret $B_\xi(-p \oplus_M x)$ as the signed distance between the origin $o$ and the horocycle $\eta(-p \oplus_M x, \xi)$ which is the unique horocycle (Helgason, 1979) through $-p \oplus_M x$ with normal $\xi$. It implies that $o$ must lie on the horocycle $\eta(-p \oplus_M x, \xi)$. Hence, points $-p \oplus_M x$ must lie on the (unique) horocycle (in purple) through the origin $o$ with normal $\xi$ (see Fig. 4(b)). It can be seen that the characterization of our hyperplanes and that of Poincaré hyperplanes are different, resulting in different formulations for the point-to-hyperplane distance.

## K LIMITATIONS OF OUR APPROACH

In our work, the distance between a point and a hyperplane is derived for all higher-rank symmetric spaces of noncompact type, and the proposed FC layers are also designed for neural networks on those spaces. However, the attention module relies on the computation of wFM which does not have a closed-form solution in the general case. To address this issue, one can consider using the Karcher algorithm (Karcher, 1977) which has proven effective in the implementation of batch normalization layers in SPD neural networks (Brooks et al., 2019). This algorithm can be computationally expensive in practice due to its iterative nature. To alleviate this challenge, the authors of Chakraborty et al. (2020) introduced an efficient wFM estimator which is worth investigating. One can also consider using the methods proposed in Lou et al. (2020). In particular, the one that relies on an exponential map reparameterization (Lezcano-Casado, 2019) can offer an effective solution to our problem.

To obtain a closed form for the point-to-hyperplane distance, one has to derive a closed form for the Busemann function on the considered spaces and Riemannian metrics which is not always trivial.

Our FC layers have memory complexity $O(d_{in}^2 d_{out})$ and time complexity $O(d_{in}^3 d_{out})$. Even though our method significantly reduces the complexity of FC layers proposed in Nguyen et al. (2024), it still does not scale well to high-dimensional input and output matrices of FC layers. Our attention block is built from these FC layers and thus suffers from the same issue.

Finally, since our approach is developed for symmetric spaces of noncompact type, it cannot be applied to those of compact type, e.g., Grassmann manifolds that are also encountered in many machine learning applications.

## L DEFINITIONS AND BASIC FACTS

### L.1 HYPERBOLIC SPACES AS SYMMETRIC SPACES OF NONCOMPACT TYPE

Here we describe the Poincaré disk model of the 2-dimensional hyperbolic space from a symmetric space perspective. Denote by $\mathbb{D} = \{x \in \mathbb{C} : \|x\| < 1\}$ the open unit disk in $\mathbb{C}$ equipped with the Riemannian metric

$$\langle u, v \rangle_x = \frac{\langle u, v \rangle}{(1 - \|x\|^2)^2},$$

where $u, v \in T_x \mathbb{D}$ are tangent vectors at $x \in \mathbb{D}$. Let $G$ be the group defined as

$$G = SU(1,1) := \left\{ \begin{bmatrix} a & b \\ \bar{b} & \bar{a} \end{bmatrix} : a, b \in \mathbb{C}, \|a\|^2 - \|b\|^2 = 1 \right\}.$$

Let $SO_m$ be the group of $m \times m$ orthogonal matrices of determinant 1. Then $\mathbb{D}$ can be identified as

$$\mathbb{D} \simeq SU(1,1)/SO_2.$$

The subgroups $K$, $A$, and $N$ in the Iwasawa decomposition $G = KAN$ and the centralizer $M$ of $A$ in $K$ are given by

$$K = \left\{ \begin{bmatrix} e^{i\theta} & 0 \\ 0 & e^{-i\theta} \end{bmatrix} : \theta \in [0, 2\pi) \right\},$$

$$A = \left\{ \begin{bmatrix} \cosh t & \sinh t \\ \sinh t & \cosh t \end{bmatrix} : t \in \mathbb{R} \right\},$$

$$N = \left\{ \begin{bmatrix} 1 + is & -is \\ is & 1 - is \end{bmatrix} : s \in \mathbb{R} \right\},$$

$$M = \left\{ \begin{bmatrix} 1 & 0 \\ 0 & 1 \end{bmatrix}, \begin{bmatrix} -1 & 0 \\ 0 & -1 \end{bmatrix} \right\}.$$

The group $G$ acts on $\mathbb{D}$ by isometries via the Möbius action defined as

$$g[x] := \frac{ax + b}{\bar{b}x + \bar{a}}.$$

This map is conformal and maps circles and lines into circles and lines.

### L.2 Pullback Euclidean Metrics

Under PEM, the Riemannian operations are given by:

$$\exp_x(u) = \phi^{-1}(\phi(x) + D_x\phi(u)),$$

$$\log_x(y) = D_{\phi(x)}\phi^{-1}(\phi(y) - \phi(x)),$$

$$\mathcal{T}_{x \to y}(u) = D_{\phi(y)}\phi^{-1} \circ D_x\phi(u),$$

where $\exp_x(.)$, $\log_x(.)$, and $\mathcal{T}_{x \to y}(.)$ are the exponential map, logarithmic map, and parallel transport, respectively.

### L.3 SPD Manifolds as Symmetric Spaces of Noncompact Type

The SPD manifold $\mathrm{Sym}_m^+$ is a differentiable manifold of dimension $m(m+1)/2$. The tangent space $T_x \mathrm{Sym}_m^+$ at point $x \in \mathrm{Sym}_m^+$ of the manifold is isomorphic to $\mathrm{Sym}_m$. The Riemannian metric is given by

$$\langle u, v \rangle_x = \mathrm{Tr}(x^{-1}ux^{-1}v),$$

where $u, v \in T_x \mathrm{Sym}_m^+$. This metric is $G$-invariant.

Let $GL_m$ be the group of $m \times m$ invertible matrices, and let $O_m$ be the group of $m \times m$ orthogonal matrices. Then $\mathrm{Sym}_m^+$ can be identified as

$$\mathrm{Sym}_m^+ \simeq GL_m/O_m.$$

Let $G = GL_m$. Then the subgroups $K$, $A$, and $N$ in the Iwasawa decomposition $G = KAN$ are given by

- $K = O_m$.
- $A$ is the subgroup of $m \times m$ diagonal matrices with positive diagonal entries.
- $N$ is the subgroup of $m \times m$ upper-triangular matrices with diagonal entries 1.

Any $g \in G$ can be written as $g = kan$ for exactly one triple $(k, a, n) \in K \times A \times N$, and the map $K \times A \times N \to G$ sending $(k, a, n)$ to $kan$ is a diffeomorphism. The centralizer $M$ of $A$ in $K$ is $M := C_K(A) := \{k \in K | ka = ak \text{ for all } a \in A\}$, which is the set of diagonal matrices with entries $\pm 1$. The (transitive) action of $G$ on $\mathrm{Sym}_m^+$ is defined as $g[x] := gxg^T$ for any $g \in G$ and $x \in \mathrm{Sym}_m^+$.

### L.4 Symmetric Spaces of Noncompact Type

Symmetric spaces of compact type and of noncompact type are interchanged by Cartan duality. Each of those can be further categorized into two classes.

Symmetric spaces of compact type have non-negative sectional curvature. The two classes of symmetric spaces of compact type are:

- Homogeneous spaces of a compact Lie group defined by an involution (class 1).
- Compact Lie groups with bi-invariant metrics (class 2).

Symmetric spaces of noncompact type have non-positive sectional curvature and are diffeomorphic to Euclidean spaces. The two classes of symmetric spaces of noncompact type are:

- Homogeneous spaces of a noncompact, noncomplex Lie group, by a maximal compact subgroup (class 3, dual to class 1).
- Homogeneous spaces of a complex Lie group by a real form (class 4, dual to class 2).

There is a correspondence between symmetric spaces of noncompact type and semisimple Lie groups with trivial centre and no compact factors: For any Lie group G with trivial centre and no

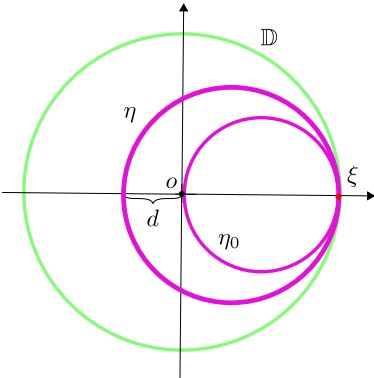

Figure 5: The boundary $\partial\mathbb{D}$ of the Poincaré disk model $\mathbb{D}$ is illustrated by the green circle. The boundary point $\xi$ is normal to both the horocycle $\eta$ and the basic horocycle $\eta_0$. The distance between the origin $o$ and the horocycle $\xi$ is $-d$.

compact factors, if we take a maximal compact subgroup $K$, then the quotient $X = G/K$ endowed with a G-invariant Riemannian metric forms a symmetric space of noncompact type.

An important invariant of a symmetric space of noncompact type is its rank. The rank of a symmetric space $X$ of noncompact type is the maximum dimension of flats in $X$ (flats in $X$ are subspaces isometric to Euclidean spaces). A symmetric space of noncompact type can be of rank one or of higher-rank.

- Rank one symmetric spaces of noncompact type: The real, complex and quaternionic hyperbolic spaces and the hyperbolic plane over the Cayley numbers.

- Higher-rank symmetric spaces of noncompact type: All symmetric spaces of noncompact type of rank greater than one. Typical examples are SPD manifolds.

Many geometric properties of rank one symmetric spaces of noncompact type and those of higher-rank are distinct. Thus, it might be challenging to extend a method designed for hyperbolic spaces to the higher-rank symmetric space setting. For instance, in the Poincaré model, the point-to-hyperplane distance resulted from geodesic projections (Chami et al., 2021) can be computed in closed-form (Ganea et al., 2018b). However, this is not the case for SPD manifolds (except those under specific Riemannian metrics, e.g. Log-Euclidean metrics) (Nguyen & Yang, 2023).

In the following, we provide several examples to illustrate some concepts reviewed in Section 3.2.

**Geometric boundary**    Some examples of geometric boundaries of symmetric spaces of noncompact type are given below:

(1) For the Poincaré disk model of the 2-dimensional hyperbolic space described in Section L.1, its geometric boundary $\partial\mathbb{D}$ is the unit circle $\partial\mathbb{D} = \{x \in \mathbb{C} : \|x\| = 1\}$ (see Fig. 5).

(2) Consider the Poincaré model $\mathbb{B}_m$ of $m$-dimensional hyperbolic geometry discussed in Section 3.1. Two geodesic rays $\delta$ and $\delta'$ in $\mathbb{B}_m$ are asymptotic if and only if $\delta(t)$ and $\delta'(t)$ converge to the same point of $\mathbb{R}^m$ as $t \to \infty$. Thus the geometric boundary $\partial\mathbb{B}_m$ of $\mathbb{B}_m$ is naturally identified with the sphere of Euclidean radius 1 centred at $\mathbf{0} \in \mathbb{R}^m$.

(3) For $\mathrm{Sym}_m^+$, its geometric boundary $\partial\,\mathrm{Sym}_m^+$ consists of the left cosets

$$\partial\,\mathrm{Sym}_m^+ := K/M := \{\xi = kM | k \in K\}.$$

(4) If X is a complete $m$-dimensional Riemannian manifold of non-positive sectional curvature, then $\partial X$ is homeomorphic to $\mathbb{S}^{m-1}$, the $(m-1)$-sphere. This is because for a given base point $x \in X$, there exists a homeomorphism which associates to each unit vector $u$ tangent to $X$ at $x$ the class of the geodesic ray $\delta$ which issues from $x$ with velocity vector $u$.

**Horocycles**    The horocycles of $\mathbb{D}$ and $\mathrm{Sym}_m^+$ can be described as follows.

(1) The geodesics in $\mathbb{D}$ are circular arcs perpendicular to the boundary $\partial\mathbb{D}$. All circular arcs perpendicular to the same point at $\partial\mathbb{D}$ can be seen as parallel lines. Thus a natural notion of horocycle is that of circle tangent to $\partial\mathbb{D}$ (see Fig. 5).

(2) Let $I_m$ be the $m \times m$ identity matrix. Then any horocycle $\eta$ can be written as $\eta = kaN[I_m]$ for some $k \in K$ and $a \in A$. In particular, it is the set of matrices having $a^2$ as diagonal matrix in the $UDU$ decomposition with respect to the $\mathbb{R}^m$-basis $\{ke_i\}_{i=1,\ldots,m}$, where $\{e_i\}_{i=1,\ldots,m}$ is the standard basis of $\mathbb{R}^m$ (Bartolucci et al., 2021).

**Composite distances**    For $\mathrm{Sym}_m^+$, the composite distance $A(x, \xi)$ from the origin $o$ to the horocycle passing through a point $x \in \mathrm{Sym}_m^+$ with normal $\xi$ is given (Sonoda et al., 2022) by

$$A(x = gK, \xi = kM) = \frac{1}{2}\log\gamma(k^T[x]),$$

where the map $\gamma : \mathrm{Sym}_m^+ \to A$ is determined by $y = v\gamma(y)v^T$ with $y \in \mathrm{Sym}_m^+$, $v \in N$.

## L.5    ANGLES

**Definition L.1.** *A comparison triangle in $E^2$ for a triple of points $(p, q, r)$ in $X$ is a triangle in the Euclidean plane with vertices $\bar{p}, \bar{q}, \bar{r}$ such that $d(p, q) = d(\bar{p}, \bar{q}), d(q, r) = d(\bar{q}, \bar{r})$, and $d(p, r) = d(\bar{p}, \bar{r})$. Such a triangle is unique up to isometry, and shall be denoted $\overline{\Delta}(p, q, r)$. The interior angle of $\overline{\Delta}(p, q, r)$ at $\bar{p}$ is called the comparison angle between $q$ and $r$ at $p$ and is denoted $\overline{\angle}_p(q, r)$. The comparison angle is well-defined provided $q$ and $r$ are both distinct from $p$.*

**Definition L.2.** *Let $\delta : [0, a] \to X$ and $\delta' : [0, a'] \to X$ be two geodesic paths with $\delta(0) = \delta'(0)$. Given $t \in (0, a]$ and $t' \in (0, a']$, we consider the comparison triangle $\overline{\Delta}(\delta(0), \delta(t), \delta'(t'))$, and the comparison angle $\overline{\angle}_{\delta(0)}(\delta(t), \delta'(t'))$. The (Alexandrov) angle or the upper angle between the geodesic paths $\delta$ and $\delta'$ is the number $\angle_{\delta,\delta'} \in [0, \pi]$ defined by:*

$$\angle_{\delta,\delta'} := \limsup_{t,t' \to 0} \overline{\angle}_{\delta(0)}(\delta(t), \delta'(t')) = \lim_{\varepsilon \to 0} \sup_{0 < t,t' < \varepsilon} \overline{\angle}_{\delta(0)}(\delta(t), \delta'(t')).$$

## L.6    BUSEMANN FUNCTIONS

**Euclidean spaces (Bridson & Häfliger, 2011)**    Let $\delta(t) = tu$ be a ray in $\mathbb{R}^m$, where $u$ is a unit vector. Then the Busemann function $B_{\xi = \delta(\infty)}(.)$ is given by

$$B_\xi(x) = -\langle x, u \rangle.$$

**SPD manifolds under Log-Euclidean framework (Bonet et al., 2023)**    Let $\delta(t) = \exp(ta)$ be a geodesic line in $\mathrm{Sym}_m^+$, where $a \in \mathrm{Sym}_m$ and $\|a\| = 1$. Then the Busemann function $B_{\xi = \delta(\infty)}(.)$ is given by

$$B_\xi(x) = -\mathrm{Tr}(a\log(x)).$$

## L.7    OPERATIONS

### L.7.1    POINCARÉ MODEL AND MÖBIUS GYROVECTOR SPACES

In the Poincaré model $\mathbb{B}_m$ of $m$-dimensional hyperbolic geometry, the logarithmic map $\log_\mathbf{0}(.)$ and exponential map $\exp_\mathbf{0}(.)$ are given as

$$\log_\mathbf{0}(x) = \tanh^{-1}(\|x\|)\frac{x}{\|x\|}, \quad \exp_\mathbf{0}(v) = \tanh(\|v\|)\frac{v}{\|v\|},$$

where $x \in \mathbb{B}_m \setminus \{\mathbf{0}\}$ and $v \in T_\mathbf{0}\mathbb{B}_m \setminus \{\mathbf{0}\}$.

The Möbius addition $\oplus_M$ is defined as

$$x \oplus_M y = \frac{(1 + 2\langle x, y \rangle + \|y\|^2)x + (1 - \|x\|^2)y}{1 + 2\langle x, y \rangle + \|x\|^2\|y\|^2},$$

where $x, y \in \mathbb{B}_m$. The Möbius subtraction $\ominus_M$ is then defined as

$$\ominus_M x = -x.$$

The Möbius scalar multiplication $\otimes_M$ is defined as

$$r \otimes_M x = \tanh(r \tanh^{-1}(\|x\|)) \frac{x}{\|x\|},$$

where $r \in \mathbb{R}$ and $x \in \mathbb{B}_m \setminus \{\mathbf{0}\}$.

The Möbius matrix-vector multiplication $\otimes_M$ is defined as

$$M \otimes_M x = \tanh \left( \frac{\|xM\|}{\|x\|} \tanh^{-1}(\|x\|) \right) \frac{xM}{\|xM\|},$$

where $x \in \mathbb{B}_m$, $M \in \mathbb{R}^{m \times m'}$, and $M \otimes_M x = \mathbf{0}$ if $xM = \mathbf{0}$. Note that we use the same notation $\otimes_M$ for the Möbius scalar multiplication and Möbius matrix-vector multiplication as in Ganea et al. (2018b).

### L.7.2 LE GYROVECTOR SPACES

For $x, y \in \mathrm{Sym}_m^+$, the binary operation $\oplus_{le}$ and inverse operation $\ominus_{le}$ are given as (Nguyen, 2022a;b)

$$x \oplus_{le} y = \exp(\log(x) + \log(y)), \quad \ominus_{le} x = x^{-1},$$

where $\exp(.)$ denotes the matrix exponential[7].

The SPD inner product is defined as

$$\langle x, y \rangle^{le} = \langle \log(x), \log(y) \rangle.$$

## M  MATHEMATICAL PROOFS

### M.1  PROOF OF PROPOSITION 4.4

*Proof.* We first recast a result from Chen et al. (2024a) (Lemma 3.5) in form of the following proposition.

**Proposition M.1.** *Let $\phi : \mathrm{Sym}_m^+ \to \mathrm{Sym}_m$ be a diffeomorphism, $p \in \mathrm{Sym}_m^+$, $a' \in T_p \mathrm{Sym}_m^+ \setminus \{\mathbf{0}\}$, and let $\mathcal{H}_{a',p}^{pb}$ be the hyperplane defined as*

$$\mathcal{H}_{a',p}^{pb} = \{x \in \mathrm{Sym}_m^+ : \langle \mathrm{Log}_p(x), a' \rangle_p^\phi = 0\},$$

*where $\langle .,. \rangle_p^\phi$ is the PEM at point $p$ as given in the definition of SPD manifolds. Then the distance $d^{pb}(x, \mathcal{H}_{a',p}^{pb})$ between a point $x \in \mathrm{Sym}_m^+$ and hyperplane $\mathcal{H}_{a',p}^{pb}$ is given by*

$$d^{pb}(x, \mathcal{H}_{a',p}^{pb}) = \frac{|\langle \phi(x) - \phi(p), D_p \phi(a') \rangle|}{\|a'\|_p^\phi},$$

*where $\|.\|_p^\phi$ is the norm induced by the Riemannian inner product $\langle .,. \rangle_p^\phi$.*

By the triangle inequality,

$$d(\delta(0), \delta(t)) - d(x, \delta(0)) \leq d(x, \delta(t)) \leq d(\delta(0), \delta(t)) + d(x, \delta(0)),$$

which gives

$$t - d(x, \delta(0)) \leq d(x, \delta(t)) \leq t + d(x, \delta(0)).$$

Thus

$$1 - \frac{d(x, \delta(0))}{2t} \leq \frac{d(x, \delta(t)) + t}{2t} \leq 1 + \frac{d(x, \delta(0))}{2t},$$

---

[7]As for function $\log(.)$, the meaning of function $\exp(.)$ should be clear from the context.

which leads to $\lim_{t\to\infty} \frac{d(x,\delta(t))+t}{2t} = 1$. Therefore

$$
\begin{aligned}
B_{\xi=\delta(\infty)}(x) &= \lim_{t\to\infty} d(x,\delta(t)) - t \\
&= \lim_{t\to\infty} \left(d(x,\delta(t)) - t\right)\frac{d(x,\delta(t))+t}{2t} \\
&= \lim_{t\to\infty} \frac{1}{2t}\left(d(x,\delta(t))^2 - t^2\right) \\
&= \lim_{t\to\infty} \frac{1}{2t}\left(\|\phi(x) - \phi(\delta(t))\|^2 - t^2\right) \\
&= \lim_{t\to\infty} \frac{1}{2t}\left(\|\phi(x)\|^2 + \|\phi(\delta(t))\|^2 - 2\langle\phi(\delta(t)),\phi(x)\rangle - t^2\right) \\
&= \lim_{t\to\infty} \frac{1}{2t}\left(\|\phi(x)\|^2 + \|ta\|^2 - 2\langle ta,\phi(x)\rangle - t^2\right) \\
&= \lim_{t\to\infty} \frac{1}{2t}\left(\|\phi(x)\|^2 - 2t\langle a,\phi(x)\rangle\right) \\
&= \lim_{t\to\infty} \frac{1}{2t}\left(\|\phi(x)\|^2\right) - \langle a,\phi(x)\rangle \\
&= -\langle a,\phi(x)\rangle.
\end{aligned}
$$

By the definitions of the binary operation $\oplus$ and inverse operation $\ominus$, we have

$$
\ominus p \oplus x = \phi^{-1}(\phi(x) - \phi(p)).
$$

Hence

$$
\|\ominus p \oplus x\|_{\mathbb{S}} = \|\phi(\ominus p \oplus x)\| = \|\phi(x) - \phi(p)\|.
$$

We then get

$$
\begin{aligned}
\bar{d}(x,\mathcal{H}_{\xi,p}) &= d(x,p).\frac{B_\xi(\ominus p \oplus x)}{\|\ominus p \oplus x\|_{\mathbb{S}}} \\
&= -d(x,p).\frac{\langle a,\phi(\ominus p \oplus x)\rangle}{\|\phi(x) - \phi(p)\|} \\
&= -\langle a,\phi(\phi^{-1}(\phi(x) - \phi(p)))\rangle \\
&= -\langle a,\phi(x) - \phi(p)\rangle.
\end{aligned}
\tag{9}
$$

From Proposition M.1,

$$
d^{pb}(x,\mathcal{H}^{pb}_{a',p}) = \frac{|\langle\phi(x) - \phi(p), D_p\phi(a')\rangle|}{\|a'\|_p^\phi} = \left|\langle\phi(x) - \phi(p), \frac{D_p\phi(a')}{\|a'\|_p^\phi}\rangle\right|,
$$

By the property of pullback metrics,

$$
\langle a_1, a_2\rangle_p^\phi = \langle D_p\phi(a_1), D_p\phi(a_2)\rangle,
$$

where $a_1, a_2 \in T_p \operatorname{Sym}_m^+$. We deduce that

$$
\left\|\frac{D_p\phi(a')}{\|a'\|_p^\phi}\right\| = 1.
$$

It can be seen that the unsigned distance $|\bar{d}(x,\mathcal{H}_{\xi,p})|$ has precisely the same form as $d^{pb}(x,\mathcal{H}^{pb}_{a',p})$.

$\square$

## M.2 PROOF OF PROPOSITION 4.8

*Proof.* To prove (i), we need a result from Kassel (2009).

**Lemma M.2.** *Let $\rho : X \to \overline{\mathfrak{a}^+}$ be the map sending $x = g[o] \in X$ to $\mu(g)$, where $g \in G$. For all $x, x' \in X$,*

$$\|\rho(x) - \rho(x')\| \leq d(x, x').$$

*Moreover, if $x, x' \in \exp(\overline{\mathfrak{a}^+})[o]$, then $d(x, x') = \|\rho(x) - \rho(x')\|$.*

Let $x = gK$, $y = hK$ where $g, h \in G$. Since the distance $d(.,.)$ is $G$-invariant, we have

$$d(x, y) = d(g^{-1}[x], g^{-1}[y])$$
$$= d(o, g^{-1}h[o]).$$

Let $g^{-1}h = kak'$ where $a \in \exp(\overline{\mathfrak{a}^+})$, $k, k' \in K$. Then

$$d(o, g^{-1}h[o]) = d(k^{-1}[o], k^{-1}kak'[o])$$
$$= d(o, a[o])$$

By Lemma M.2,

$$d(o, a[o]) = \|\rho(o) - \rho(a[o])\|$$
$$= \|\mu(a)\|$$
$$= \|\mu(g^{-1}h)\|.$$

Note that

$$\| \ominus x \oplus y\|_{\mathbb{S}} = \|g^{-1}hK\|_{\mathbb{S}}$$
$$= \sqrt{\langle g^{-1}hK, g^{-1}hK\rangle_{\mathbb{S}}}$$
$$= \sqrt{\langle \mu(g^{-1}h), \mu(g^{-1}h)\rangle}$$
$$= \|\mu(g^{-1}h)\|.$$

Therefore

$$\| \ominus x \oplus y\|_{\mathbb{S}} = d(x, y).$$

To prove (ii), note that for any $k \in K$, we have $k[x] = kgK = kk_1a_1n_1K = k_2a_1n_1K$ where $g = k_1a_1n_1$, $k_2 = kk_1$, $k_1 \in K$, $a_1 \in A$, $n_1 \in N$. Thus $\mu(kg) = \mu(g)$. Similarly, we deduce that $\mu(kh) = \mu(h)$. Therefore

$$\langle k[x], k[y]\rangle_{\mathbb{S}} = \langle \mu(kg), \mu(kh)\rangle$$
$$= \langle \mu(g), \mu(h)\rangle$$
$$= \langle x, y\rangle_{\mathbb{S}}.$$

$\square$

## M.3  PROOF OF PROPOSITION 4.9

*Proof.* We first recast a result from Bridson & Häfliger (2011) (Lemma 10.26) in form of the following lemma.

**Lemma M.3.** *Let $X$ be a symmetric space of noncompact type and let $G$ be a group acting by isometries on $X$. Suppose that $h \in G$ leaves invariant a geodesic line $\delta : \mathbb{R} \to X$ and that $h[\delta(t)] = \delta(t + c)$ where $c > 0$. Let $x_0 = \delta(0)$ and let $N \subset G$ be the set of elements $g \in G$ such that $h^{-j}gh^j[x_0] \to x_0$ as $j \to \infty$. Then $N$ fixes $\delta(\infty) \in \partial X$ and leaves invariant the Busemann function associated to $\delta$.*

Let $\delta(t) = k\exp(ta)K$, $h = \exp(a) \in G$, where $a \in \mathfrak{a}$, $\|a\| = 1$, $k \in K$. Setting $\delta'(t) = k^{-1}\delta(t)$. Then

$$h[\delta'(t)] = \exp(a + ta)K$$
$$= k^{-1}\delta(t + 1)$$
$$= \delta'(t + 1).$$

Since the distance $d(.)$ is G-invariant, for any $x \in X$, we have

$$d(x, \delta(t)) = d(k^{-1}[x], \delta'(t)).$$

Let $g \in G$ be such that $k^{-1}[x] = gK$, and let $g = n_1 \exp A(g) k_1$ where $n_1 \in N$, $k_1 \in K$[8]. For any $n \in N$, $h^{-j} n h^j[o] \to o$ as $j \to \infty$. By Lemma M.3 ($c = 1$), we deduce that $B_{\xi'=\delta'(\infty)}(k^{-1}[x]) = B_{\xi'=\delta'(\infty)}(n_1^{-1} k^{-1}[x])$. Hence

$$d(k^{-1}[x], \delta'(t))^2 = d(n_1^{-1} k^{-1}[x], \delta'(t))^2.$$

We thus have the following chain of equations

$$
\begin{aligned}
d(k^{-1}[x], \delta'(t))^2 &= d(n_1 \exp A(g) K, \delta'(t))^2 \\
&= d(\exp A(g) K, \delta'(t))^2 \\
&= \langle A(g) - ta, A(g) - ta \rangle \\
&= \langle A(g), A(g) \rangle - 2t\langle a, A(g) \rangle + t^2.
\end{aligned}
\tag{10}
$$

By the triangle inequality,

$$d(\delta'(0), \delta'(t)) - d(k^{-1}[x], \delta'(0)) \le d(k^{-1}[x], \delta'(t)) \le d(\delta'(0), \delta'(t)) + d(k^{-1}[x], \delta'(0)),$$

which gives

$$t - d(k^{-1}[x], \delta'(0)) \le d(k^{-1}[x], \delta'(t)) \le t + d(k^{-1}[x], \delta'(0)).$$

Thus

$$1 - \frac{d(k^{-1}[x], \delta'(0))}{2t} \le \frac{d(k^{-1}[x], \delta'(t)) + t}{2t} \le 1 + \frac{d(k^{-1}[x], \delta'(0))}{2t},$$

which results in $\lim_{t\to\infty} \frac{d(k^{-1}[x],\delta'(t))+t}{2t} = 1$. Therefore

$$
\begin{aligned}
B_{\xi=\delta(\infty)}(x) &= \lim_{t\to\infty} d(x, \delta(t)) - t \\
&= \lim_{t\to\infty} d(k^{-1}[x], \delta'(t)) - t \\
&= \lim_{t\to\infty} \left( d(k^{-1}[x], \delta'(t)) - t \right) \frac{d(k^{-1}[x], \delta'(t)) + t}{2t} \\
&= \lim_{t\to\infty} \frac{1}{2t} \left( d(k^{-1}[x], \delta'(t))^2 - t^2 \right).
\end{aligned}
$$

Using Eq. (10), we get

$$
\begin{aligned}
B_\xi(x) &= \lim_{t\to\infty} \frac{1}{2t} \left( \langle A(g), A(g) \rangle - 2t\langle a, A(g) \rangle \right) \\
&= -\langle a, A(g) \rangle \\
&= \langle a, H(g^{-1}) \rangle,
\end{aligned}
$$

which concludes Proposition 4.9.

$\square$

## M.4    PROOF OF COROLLARY 4.10

*Proof.* We have

$$
\begin{aligned}
\bar{d}(x, \mathcal{H}_{\xi,p}) &= d(x, p) \cdot \frac{B_\xi(\ominus p \oplus x)}{\|\ominus p \oplus x\|_{\mathbb{S}}} \\
&= d(x, p) \cdot \frac{B_\xi(h^{-1} g K)}{\|\ominus p \oplus x\|_{\mathbb{S}}}.
\end{aligned}
$$

---

[8]We use the same notation $A$ for the composite distance from the origin $o$ to a horocycle as in Helgason (1994).

Note that $k^{-1}[h^{-1}gK] = k^{-1}h^{-1}gK$. By Propositions 4.8 and 4.9,

$$d(x,p).\frac{B_\xi(h^{-1}gK)}{\|\ominus p \oplus x\|_{\mathbb{S}}} = \langle a, H(g^{-1}hk)\rangle,$$

which concludes Corollary 4.10.

$\square$

## M.5 PROOF OF PROPOSITION 5.1

*Proof.* We first recast a result from Bridson & Häfliger (2011) (Proposition 9.8) in form of the following proposition.

**Proposition M.4.** *Let $X$ be a symmetric space of noncompact type with basepoint $x_0$. Let $\xi, \xi' \in \partial X$ and let $\delta, \delta'$ be geodesic rays with $\delta(0) = \delta'(0) = x_0$, $\delta(\infty) = \xi$ and $\delta'(\infty) = \xi'$. Then*

$$2\sin(\angle(\xi,\xi')/2) = \lim_{t\to\infty} \frac{1}{t}d(\delta(t),\delta'(t)).$$

For any $t \in [0,\infty)$, we have that

$$\begin{aligned}
d(\delta(t),\delta'(t)) &= d(\exp(ta)K, \exp(ta')K) \\
&= \|t(a-a')\| \\
&= \sqrt{2}t.
\end{aligned}$$

By Proposition M.4,

$$\begin{aligned}
2\sin(\angle(\xi,\xi')/2) &= \lim_{t\to\infty} \frac{1}{t}d(\delta(t),\delta'(t)) \\
&= \sqrt{2}.
\end{aligned}$$

We thus deduce that $\angle(\xi,\xi') = \frac{\pi}{2}$.

$\square$

## M.6 PROOF OF PROPOSITION 5.2

*Proof.* Let $(e_j)_{j=1,\ldots,m}$ be the standard basis of $\mathbb{R}^m$, and let $\tilde{\xi}_j = \tilde{\delta}_j(\infty)$ where $\tilde{\delta}_j(t) = \exp(te_j)K, j = 1,\ldots,m$ be geodesic rays. Then for $y = gK \in X$ and any $j \in \{1,\ldots,m\}$,

$$v_j(x) = \bar{d}(y, \mathcal{H}_{\tilde{\xi}_j,K}) = \langle e_j, H(g^{-1})\rangle = H(g^{-1})[j],$$

where $H(g^{-1})[j]$ denotes the $j$-th dimension of $H(g^{-1})$. Thus

$$H(g^{-1}) = [v_1(x)\ldots v_m(x)]^T.$$

Note that $g = n\exp(-H(g^{-1}))k$ with $n \in N$ and $k \in K$. Therefore

$$g = n\exp([-v_1(x)\ldots - v_m(x)])k,$$

which leads to

$$y = n\exp([-v_1(x)\ldots - v_m(x)])K.$$

$\square$

