# OpenReview forum: "Neural networks on Symmetric Spaces of Noncompact Type"
_ICLR.cc/2025/Conference — ICLR 2025 Poster_

### Official Review · Reviewer_yrjK · 2024-10-30

**Soundness:** 2
**Presentation:** 1
**Contribution:** 3
**Rating:** 6
**Confidence:** 2

**Summary:**

The paper presents a new formulation for the distance between a point and a hyperplane in symmetric spaces, i.e., spaces that generalize the commonly used hyperbolic spaces and the manifold of SPD matrices. Based on this distance, new FC and attention components of neural networks on such symmetric spaces are proposed. These neural networks are then demonstrated in the context of hyperbolic spaces and the SPD manifold, showing an advantage in performance compared to other existing methods.

**Strengths:**

- **Solid Mathematical Foundation**: The paper introduces new definitions grounded in a well-developed mathematical framework.
- **New Neural Network Architecture**: The authors propose a new neural network design, including fully connected (FC) layers and an attention mechanism, that operate on symmetric spaces, extending the current capabilities of neural networks in non-Euclidean geometries.

**Weaknesses:**

- **Presentation and Organization**: The paper's structure is fragmented, making it challenging to follow and fully appreciate its main contributions. Specifically:
    - **Sections 2 and 3** contain too many short, disconnected paragraphs or subsections that do not establish a cohesive narrative or logical organization.
    - **Section 4** would benefit from being divided into two distinct parts: one focused on the mathematical framework and the other on the proposed neural network. Since the neural network is the primary outcome, separating these would clarify the focus.
    - Additional minor structural suggestions:
        - **Section 4.3**: Consider renaming the section titled "Examples" to make its purpose clearer.
        - **Figure 2(b)**: Rotating this figure by 90 degrees could improve readability.

- **Experimental Results**: Some limitations in the experimental study reduce the impact of the findings:
    - **Section 5.1**: There are many other datasets with a clearer hierarchical structure that typically benefit more from hyperbolic space representation. For example, gene expression and word-document datasets are often considered as benchmarks of embedding in hyperbolic space. Therefore, the choice of CIFAR-10 and CIFAR-100 as benchmarks is questionable and should be explained, especially considering the marginal improvements reported in Table 2.
    - **Section 5.2**: There are multiple differences between the proposed methods and the baselines, with several moving parts. It remains unclear which component contributes most to the observed (marginal) improvement. Consider conducting an ablation study to isolate the effects of the different components. This would help clarify which aspects are most responsible for the performance improvements.
    - **Runtime Analysis**: Since the accuracy improvements are relatively small, including runtime metrics alongside accuracy would provide a fuller perspective on the method's practical benefits.

- **Limitations Statement**: The limitation statement at the end is very specific and could be extended to cover broader aspects of the proposed framework, e.g., scalability and practical applicability.

**Questions:**

- Could the authors provide clarification on the choice of CIFAR-10 and CIFAR-100 datasets in light of the results and the available alternative datasets with stronger hierarchical structures?
- Please explain why more hierarchical datasets were not included.

---

> ### Author Response · Authors · 2024-11-21
>
> We thank the reviewer for the positive feedbacks.
>
> Regarding Section 3, we propose to add more descriptions of the concepts to the Appendix, and use the Poincaré disk model for graphical illustrations of those concepts (e.g., horocycles, composite distances, Busemann functions, etc).
>
> Regarding Section 4, we propose to change section 4.5 (Neural networks on symmetric spaces) to Section 5. Sections 4.5.1 (FC layers) and 4.5.2 (Attention mechanism) become Sections 5.1 and 5.2, respectively. The experiment section changes accordingly.
>
> We propose to rename the title of Section 4.3 as ``Point-to-hyperplane Distances on Hyperbolic Spaces and SPD Manifolds".
>
> In Section 5.1, our motivations for using CIFAR-10 and CIFAR-100 datasets are (1) we already perform experiments on natural language inference tasks with datasets having hierarchical structures (please refer to Appendix C). Those datasets are used in the work of Ganea et al., 2018b that proposes the g-distance. Thus, comparing our distance against theirs on those datasets would validate the effectiveness of our method;
> and (2) it would be interesting to evaluate our method on computer vision tasks to show a broad range of applications of our method. In particular, recent works (van Spengler et al., 2023; Bdeir et al., 2024) show that deep hybrid HNNs have good capability of representing complex structures in image data.
>
> It is important to note that our main goal for the experiments in Section 5.1 is to demonstrate the generality of our method, i.e., our distance derived from a general formulation of the point-to-hyperplane distance is able to achieve competitive performance against the g-distance which is specifically constructed for hyperbolic spaces. To further investigate the generality of our method, we are performing experiments with some datasets as suggested and we shall report results of these experiments when they are done.
>
> Regarding Section 5.2, we are conducting some ablation studies and shall report results of these studies when they are done.
>
> The computation times of Euclidean-Poincaré-H, Euclidean-Poincaré-G and Euclidean-Poincaré-B for image classification experiments are
> presented in Table A. It can be observed that Euclidean-Poincaré-B has high computational costs compared to its competitors, while
> Euclidean-Poincaré-H is the fastest method among the three methods.
>
> | Method              | Euclidean-Poincaré-H | Euclidean-Poincaré-G | Euclidean-Poincaré-B |
> |-----------------------|--------------------------------|---------------------------------|--------------------------------|
> |   CIFAR-10         |    52    |     57    |     59     |
> |   CIFAR-100       |    90    |    114   |     118   |
> Table A: Computation times (seconds) per epoch for image classification experiments (measured on a Quadro RTX 8000 GPU).
>
> We do not investigate further on EEG signal classification experiments since the performance gaps between
> our method and state-of-the-art methods are clear.
>
> Some limitations of our approach are:
> - Since our approach is developed for symmetric spaces of noncompact type, it cannot be applied to those of compact type, e.g., Grassmann manifolds that are also encountered in many machine learning applications.
> - To obtain a closed form for the point-to-hyperplane distance, one has to derive a closed form for the Busemann function on the considered spaces and Riemannian metrics which is not always trivial.
> - Our FC layers have memory complexity $O(d_{in}^2d_{out})$ and time complexity $O(d_{in}^3d_{out})$. Even though our method
> significantly reduces the complexity of FC layers proposed in Nguyen et al. (2024), it still does not scale well to high-dimensional input and output matrices of FC layers. Our attention block is built from these FC layers and thus suffers from the same issue.

---

> > ### Author Response · Authors · 2024-11-23
> >
> > As requested, we performed node classification experiments on Disease [J], Airport [J], Pubmed [K], and Cora [L] datasets. These experiments can be added to Appendix on reviewer’s demande and source code will be made available upon acceptance of the paper. Since we only aim to compare the h-distance, g-distance, and b-distance, we do not seek state-of-the-art results in these experiments. We also would like to mention that like our experiments on image classification, image generation, and natural language inference, our goal is to show the generality of our method.
> >
> > **Disease dataset** It is the transductive variant of a dataset created by simulating the SIR disease spreading model where the label of a node is whether the node was infected or not, and node features indicate the susceptibility to the disease.
> >
> > **Airport dataset** It is a flight network dataset from OpenFlights.org where nodes represent airports, edges represent the airline Routes, and node labels are the populations of the country where the airport belongs.
> >
> > **Pubmed dataset** It is a standard benchmark describing citation networks where nodes represent scientific papers in the area of medicine, edges are citations between them, and node labels are academic (sub)areas.
> >
> > **Cora dataset** It is a citation network where nodes represent scientific papers in the area of machine learning, edges are citations between them, and node labels are academic (sub)areas.
> >
> > The statistics of the four datasets are summarized in Table B.
> >
> > | Dataset | \#Nodes | \#Classes | \#Features |
> > |-----------------------|-------------|-------------|------------|
> > |  Disease  |    1044    |   2  |   1000    |
> > |  Airport  |    3188    |   4  |   4    |
> > |  Pubmed  |    19717    |   3  |   500    |
> > |  Cora  |    2708    |   7  |   1433    |
> >
> > Table B: Description of the datasets for node classification.
> >
> > Following [J], we use 30/10/60 percent splits for Disease dataset, 70/15/15 percent splits for Airport dataset, and standard splits with 20 train examples per class for Pubmed and Cora datasets.
> >
> > **Network architecture** We use the HGCN architecture in [J] which consists of a graph convolutional network (GCN) and a MLR layer for classification. Both the GCN and the MLR layer are built on the Poincaré ball.
> >
> > **Hyperparameters** We follow the experimental settings in [J]. The number of epochs is set to 5000. We use early stopping based on validation set performance with a patience of 100 epochs. The learning rate is set to $1e-2$. The weight decays for Diease and Airport datasets are set to 0. The weight decays for Pubmed and Cora datasets are set to $1e-4$ and $1e-3$, respectively. The number of dimensions is set to 16. The number of convolutional layers in the GCN is set to 3.
> >
> > **Optimization and evaluation** The network is implemented in Pytorch and is trained using cross-entropy loss and Adam optimizer. Results are averaged over 10 random parameter initializations on the final test set.
> >
> > **Results** Table C shows results of our experiments. It can be observed that the b-distance gives the best performance in most cases. It can also be observed that the h-distance is significantly outperformed by the g-distance and b-distance on more hyperbolic datasets, i.e., Disease and Airport datasets. For those datasets, the b-distance achieves the best performances, demonstrating its effectiveness for datasets with strong hierarchical structures. Furthermore, in all cases, the b-distance achieves the lowest standard deviations, suggesting that it can offer stable results in the considered application.
> >
> > | Dataset | Disease | Airport | Pubmed | Cora |
> > |-----------------------|-------------|-------------|------------|-------------------|
> > |  Hyperbolicity $\delta$  |    $\delta=0$    |  $\delta=1$  | $\delta=3.5$  |  $\delta=11$  |
> > |  HGCN-H  |   85.67 ± 2.58    |  69.82 ± 2.08  | 75.26 ± 1.82  |  77.09 ± 2.02  |
> > |  HGCN-G  |   88.98 ± 1.96   |  84.78 ± 1.48  | **76.02 ± 1.09**  |  77.47 ± 1.15  |
> > |  HGCN-B (Ours)  |   **89.05 ± 0.78**   |  **85.04 ± 0.97**  | 75.89 ± 0.78  |  **77.90 ± 1.00**  |
> >
> > Table C: Results of HGCN models for node classification. HGCN-H, HGCN-G, and HGCN-B are built on the h-distance, g-distance, and b-distance, respectively. A lower hyperbolicity value δ means more hyperbolic.
> >
> > **References**
> >
> > [J] Ines Chami, Rex Ying, Christopher R, and Jure Leskovec. Hyperbolic Graph Convolutional Neural Networks. CoRR, abs/1910.12933, 2019. URL https://arxiv.org/abs/1910.12933.
> >
> > [K] Galileo Namata, Ben London, Lise Getoor, and Bert Huang. Query-driven Active Surveying for Collective Classification. In Workshop on Mining and Learning with Graphs, 2012.
> >
> > [L] Prithviraj Sen, Galileo Mark Namata, Mustafa Bilgic, Lise Getoor, Brian Gallagher, and Tina
> > Eliassi-Rad. Collective Classification in Network Data. AI Magazine, 29(3):93–106, 2008.

---

> > > ### Comment · Reviewer_yrjK · 2024-11-25
> > >
> > > I would like to thank the authors for their thorough and detailed responses. Most of my concerns have been adequately addressed, and I am pleased to recommend this paper for acceptance.

---

> ### Author Response · Authors · 2024-11-25
>
> We thank the reviewer for the quick response and the positive feedback.
>
> We woud like to provide results of our ablation study. Here we will focus on the effectiveness of the FC layer and attention module, since those are the practical contributions of our work. To this end, we evaluate the performance of AttSymSpd-GI in two cases:
> - The attention module is removed from the network. The resulting network is called CovNet. This allows us to validate the contribution of the attention module.
> - The FC layers in the attention module are replaced with Bimap layers [M]. Bimap layers are referred to as FC convolution-like layers and arguably the most commonly used analogs of FC layers in SPD neural networks. The resulting network is called AttSymSpd-GI-Bimap.
>
> Table D reports results of our experiments. It can be observed that both the building blocks are effective. In particular, the use of attention module leads to more than 5\% improvement in mean accuracy, and the network based on our FC layers outperforms the one based on Bimap layers by more than 2\% in terms of mean accuracy.
>
> | Method        | CovNet                       | AttSymSpd-GI-Bimap | AttSymSpd-GI        |
> |-----------------|-------------------------------|------------------------------|---------------------------|
> |                     |    73.04 $\pm$ 6.34    |   75.82 $\pm$ 5.1        |   **78.08 $\pm$ 4.8**  |
> Table D: Effectiveness of the proposed FC layer and attention module on BCIC-IV-2a dataset.
>
> **Reference**
>
> [M] Zhiwu Huang and Luc Van Gool. A Riemannian Network for SPD Matrix Learning. In AAAI, pp. 2036-2042, 2017.

---

### Official Review · Reviewer_75bJ · 2024-11-02

**Soundness:** 3
**Presentation:** 2
**Contribution:** 3
**Rating:** 6
**Confidence:** 3

**Summary:**

The paper presents a new method for defining neural networks on symmetric spaces of noncompact type. The method is derived by deriving consider how hyperplanes are defined in such spaces and how point-to-hyperplane distance can be computed. This follows from expressing inner products via Busemann functions. This is then used for defining neural networks be generalizing the observation that affine functions in Euclidean space can be expressed as a function of a point-to-hyperplane distance. By replacing quantities in Euclidean space with their symmetric space counter parts, linear layers are defined to define neural networks on these spaces.

**Strengths:**

- The approach provide a novel generalization of defining neural networks in the more general symmetric space of noncompact type. It is very appealing that the approach can be utilized on several types manifolds (Section 4.3)
- The paper provides is mostly written well to explain the technical background of the material (caveat below).
- Experimental results seem promising.

**Weaknesses:**

- The connection between the proposed approach and previous ones are not exactly clear. Mostly in how / why the are different (see Questions)
- I think the narrative of eventually defining the FC layers in section 4.5 (and the attention mechanism) could be improved. Particularly, the I feel like the connection of expressing affine functions via point-to-hyperplane distances should be further elaborated (L396-404)

**Questions:**

Questions / Remarks:

1. What are the specific connection between the proposed formulation versus the previous approaches presented in Table 1. I may be incorrect, but my understanding is that Ganea et al., 2018b is specialized for Hyperbolic spaces; and the b-distance approach is the Busemann function specialized to Hyperbolic spaces via Section 4.3 / Corollary 4.3. Is there a deeper reason why the point-to-hyperplane distances would not reduce to be the same? A reason for this question is that my initial perception was that the proposed look into symmetric spaces generalized the hyperbolic space, and thus when specializing to hyperbolic spaces one should obtain the same distance function. In summary, it would be great if you could provide a detailed summary / comparison for why the distances obtained in Ganea et al. 2018b differs from those obtained through your Busemann function approach.

2. Unsure if it is the original citation for the technique of affine maps as point-to-hyperplane distance functions, but Ganea et al., 2018b cites "Hyperplane margin classifiers on the multinomial manifold" by Lebanon & Lafferty. To this end, it would be useful to add this citation to Section 2.1 and perhaps further elaborate on the historical development of using point-to-hyperplane distances for affine maps.

Typos / Minor Mistakes:
 - Definition of hyperbolic distance is incorrect on Line 106. There is a type in the inner term's denominator. It should be "$(1 - \Vert x \Vert^2)$, the squared is in the wrong position.
 - wFM used on Line 470 before defined (in appendix Line 1192).
 - Missing "." punctuation on Line 425.

---

> ### Author Response · Authors · 2024-11-21
>
> We thank the reviewer for the positive feedbacks.
>
> L396-404 can be further elaborated as follows.
> An FC layer can be described by the following linear transformation:
> \begin{equation}
> y = ax - b,
> \end{equation}
> where $a \in \mathbb{R}^{m \times m'}$, $x \in \mathbb{R}^{m'}$, and $y,b \in \mathbb{R}^{m}$. The above equation can be rewritten as a system of equations, each for one dimension, i.e., the $j$-th dimension $y_j,j=1\ldots,m$ of the output $y$ is given as
> \begin{equation*}
> y_j = \langle x,a_j \rangle - b_j,
> \end{equation*}
> where $a_j \in \mathbb{R}^{m'},b_j \in \mathbb{R}$. Let $\tilde{\xi}\_j \in \partial X$ and let $\mathcal{H}\_{\tilde{\xi}\_j,K}$
> be the hyperplane that contains the origin (i.e., $K$) and is orthonormal to the $j$-th axis of the output space. Then $y_j$ can be interpreted as the signed distance $\bar{d}(y,\mathcal{H}\_{\tilde{\xi}\_j,K})$ from the output $y$ to hyperplane $\mathcal{H}\_{\tilde{\xi}\_j,K}$. We thus have
> \begin{equation*}
> \bar{d}(y,\mathcal{H}\_{\tilde{\xi}\_j,K}) = \langle x,a_j \rangle - b_j.
> \end{equation*}
>
> From the definition of hyperplanes on symmetric spaces (Definition 4.1), we can write the expression $\langle x,a_j \rangle - b_j$ as $B_{\xi_j}(\ominus p_j \oplus x)$, where $p_j \in X$ and $\xi_j \in \partial X$. Therefore
> \begin{equation}
> \bar{d}(y,\mathcal{H}\_{\tilde{\xi}\_j,K}) = B_{\xi_j}(\ominus p_j \oplus x),
> \end{equation}
> for $j=1,\ldots,m$.
>
> Regarding the difference between the b-distance and g-distance, please refer to Appendix F of the paper. We updated the paper with Appendix F since we could not upload a separate pdf file for Fig. 4. Appendix F can be removed or kept on the reviewer's demande.
>
> We shall add the paper ``Hyperplane margin classifiers on the multinomial manifold" to Section 2.1 and cite it in Appendix A.1.2 (the transformation of Eq. (6) to Eq. (7) is originally proposed in that paper). All the mistakes will be corrected in a new version of the paper.

---

> > ### Comment · Reviewer_75bJ · 2024-11-25
> >
> > Thank you for the clarification.
> >
> > I believe that the elaboration of L396-404 would be a useful addition to the paper.
> >
> > Regarding the additional Appendix F, I also found this very useful in answering my question. In particular, Figure 4 clarifies much of my confusion -- which I believe, in summary, is that the Poincare hyperplane and the proposed hyperplane are fundamentally different definitions. I would definitely recommend adding this (at least the image) to the main text, space permitting, and perhaps even additionally adding the "h-distance" if it visually makes sense (maybe as an extension of Table 1).
> >
> > My concerns have been addressed and I am overall positive about the paper.

---

> > > ### Author Response · Authors · 2024-11-25
> > >
> > > We thank the reviewer for the positive feedback and the suggestions for improving the exposition of our paper.

---

### Official Review · Reviewer_dL7j · 2024-11-04

**Soundness:** 3
**Presentation:** 3
**Contribution:** 3
**Rating:** 6
**Confidence:** 2

**Summary:**

This paper introduces a framework for calculating point-to-hyperplane distances within symmetric spaces of noncompact type. Building upon this theoretical foundation, the authors propose novel manifold learning blocks tailored for neural networks, particularly designing fully connected (FC) layers and an attention mechanism applicable to these spaces. The paper demonstrates the effectiveness of this approach through numerical experiments, particularly on EEG classification tasks.

**Strengths:**

1. The work presents a well-constructed theoretical basis by generalizing point-to-hyperplane distance formulations on symmetric spaces of noncompact type, encompassing both hyperbolic and SPD manifolds. This unified approach is a notable advancement that addresses the limitations in existing methodologies which often focus on narrower manifold types (e.g., Nguyen & Yang, 2023). The paper’s theoretical contribution strengthens its foundation, offering a comprehensive framework applicable across various symmetric spaces, potentially enhancing applications in machine learning on non-Euclidean geometries .
2. The experimental results, particularly on EEG datasets, demonstrate the approach’s capability. Despite minor performance gains, the proposed model achieves competitive accuracy, and in some cases, it outperforms existing methods such as EEG-TCNet, Graph-CSPNet, and MBEEGSE. This highlights the framework’s potential for real-world applications in EEG signal processing.

**Weaknesses:**

Sections 4.5.1 and 4.5.2, which are the core practical contributions of this work, are difficult to follow. While the theoretical sections are clearly presented, the implementation of the proposed FC layers and attention mechanism in symmetric spaces feels briefly discussed and lacks an intuitive explanation. A more thorough discussion, with a step-by-step breakdown or additional illustrative examples, would greatly improve accessibility and clarity.

**Questions:**

1. In Line 143, Add references following “Iwasawa decomposition of G” to provide readers with foundational context.
2. In Line 233, Correct “\(\Vert\ldot\Vert\)” to “\Vert\cdot\Vert” for accuracy in notation.
3. In Corollary 4.3, There seems to be an extraneous dot in the formula.
4. In Definition 4.5, Shouldn’t the definition of “addition” assume an abelian group structure for coherence with traditional addition in symmetric spaces?
5. In Definition 4.7, The formula for “ g = k \exp(\mu) ” in line 342 could benefit from an intuitive explanation.
6. In Proposition 4.12, The construction of the FC layer from this proposition is not immediately intuitive. Adding a diagram or further expanding on its practical implications would help readers grasp the proposed structure better.
7.  Is there complexity analysis for the two proposed blocks. such as in the supp.?

---

> ### Author Response · Authors · 2024-11-21
>
> We thank the reviewer for the positive feedbacks.
>
> Some references for Iwasawa decomposition are [F, G]. [F] provides the general theory of Iwasawa decomposition, while [G] proposes a method to compute Iwasawa decomposition on the spaces of interest. We shall add them in line 143 as suggested. We shall also correct errors in line 233 and Corollary 4.3.
>
> There are advantages and disadvantages of assuming an abelian group structure when defining the addition. In addition to the group axioms, if the commutativity of the addition also holds, then extensions of traditional machine learning models to Riemannian manifolds based on that addition may inherit good properties of the original models (from both theoretical and practical perspectives). However, making such an assumption would limit the applicability of the approach. In many cases, a relaxation of the commutativity, e.g., the gyrocommutativity ($x \oplus y = \operatorname{gyr}[x,y](y \oplus x)$ where $\operatorname{gyr}[.,.]$ is a gyroautomorphism) [H], can lead to competitive or even better performance [I]. Our work also shows that the proposed non-commutative addition can achieve good performance.
>
> In Definition 4.7, the formula $g = k \exp(\mu(g)) k'$ with $g \in G$ and $k,k' \in K$ follows from the Cartan decomposition
> $G = K (\exp \bar{\mathfrak{a}^+}) K$ [F]. This decomposition states that any $g \in G$ can be expressed as
> $g = k \exp(\mu(g)) k'$ for some $k,k' \in K$ and a unique $\mu(g) \in \bar{\mathfrak{a}^+}$.
> The map $\mu(.)$ is a continuous, proper, surjective map to the closed
> Weyl chamber $\bar{\mathfrak{a}^+}$ [F]. It has a nice property, i.e., the norm $\\|\mu(.)\\|$ is equal to the Riemannian distance function.
>
> Regarding the constructions of the FC layer and attention module, please refer to our general response (we also explain the backpropagation procedures requested by Reviewer dvJ8).
>
> Complexity analysis: Here we analyze the complexity of the two blocks with $G$-invariant metrics on SPD manifolds.  Let $d_{in}$ and $d_{out}$ be the dimensions of input and output matrices of an FC layer, respectively. Let $l$ be the number of input matrices of the attention block.
> - FC layer: It has memory complexity $O(d_{in}^2d_{out})$ and time complexity $O(d_{in}^3d_{out})$.
> - Attention layer: It has memory complexity $O(3d_{in}^2d_{out})$. The time complexity of the FC layers is $O(3ld_{in}^3d_{out})$.
> Both the "Riemannian distance" and "Midpoint operation" blocks have time complexity $O(l^2d_{out}^3)$.
>
>
> **References**
>
> [F] S. Helgason. Differential Geometry, Lie Groups, and Symmetric Spaces. ISSN. Elsevier Science, 1979.
>
> [G] P. Sawyer. Computing the Iwasawa decomposition of the classical Lie groups of noncompact type using the QR decomposition. Linear Algebra and its Applications, 493:573-579, 2016.
>
> [H] A. A. Ungar. Analytic Hyperbolic Geometry in N Dimensions: An Introduction. CRC Press, 2014.
>
> [I] X. S. Nguyen and S. Yang. Building Neural Networks on Matrix Manifolds: A Gyrovector Space Approach. In ICML, pp. 26031–26062, 2023.

---

> > ### Comment · Reviewer_dL7j · 2024-11-26
> >
> > I greatly appreciate the authors’ detailed response and have no further questions at this time.

---

> > > ### Author Response · Authors · 2024-12-01
> > >
> > > We thank the reviewer for the positive feedback and the insightful comments on our paper.

---

### Official Review · Reviewer_dvJ8 · 2024-11-07

**Soundness:** 3
**Presentation:** 2
**Contribution:** 3
**Rating:** 6
**Confidence:** 3

**Summary:**

This paper proposes neural network structures on symmetric spaces of noncompact types, such as hyperbolic spaces or symmetric positive-definite (SPD) matrix manifolds. For this purpose, an expression for the distance between a point and a hyperplane is derived and utilized to generalize the Euclidean fully connected (FC) layer and attention layer. The proposed neural network is applied to various problems to demonstrate its performance advantages.

**Strengths:**

- The proposed ideas of generalizing FC and attention layer to symmetric spaces are novel and seem reasonable.
- They are also general enough to incorporate both hyperbolic spaces and SPD manifolds, which are non-Euclidean spaces of interest and frequent use in ML.
- Most mathematical derivations seem rigorous (I could not follow all the details).
- The paper made a great effort to show the empirical benefits of the proposed neural network by considering diverse benchmarks.

**Weaknesses:**

- Understanding this paper requires a solid background in the geometry of symmetric spaces, and Section 3.2, in particular, is highly abstract and challenging to follow. This complexity may reduce the paper’s accessibility for a broader machine learning audience. Maybe incorporating the materials about decomposition equations from Appendix G.1 to G.3 in the manuscript, along with an explanation of their significance, would improve readability.
- The methods to forward propagate the FC layer and attention layer are explained, but it is not clearly defined what the inputs, outputs, dimensionalities, and trainable parameters of each layer are. Including equations or diagrams that summarize these details would enhance clarity. Additionally, there is limited discussion on the backward propagation process in each layer, particularly regarding gradient calculations. It would be helpful to specify whether gradient computation is feasible and, if so, describe the method.
- The rationale for considering both PEM and G-invariant metrics, as well as the differences between using each metric is not sufficiently discussed. A brief comparison of PEM and G-invariant metrics, with an overview of their respective strengths and limitations within the proposed neural network architecture, would help readers understand the motivation for including both metrics and their potential effects on network performance.

**Questions:**

Please refer to the weaknesses.

---

> ### Author Response · Authors · 2024-11-21
>
> We thank the reviewer for the positive feedbacks.
>
> Regarding the implementations of the FC layer, attention module, and the backpropagation procedures for these building blocks,
> please refer to our general response.
>
> The motivation for using $G$-invariant metrics is that such metrics always exist on our considered spaces (e.g.,
> those induced from the Killing form [B]). As a result, the framework proposed in Section 4.4 is valid for all those spaces.
> Since Busemann functions associated with $G$-invariant metrics are determined (see the proof of Proposition 4.9) by Busemann functions on
> the maximal abelian $A$ of $G$ (recall that $G = KAN$) which has much lower dimension than $X$,
> a major advantage of $G$-invariant metrics compared to PEM metrics is that the former allow FC layers to scale better
> to high-dimensional input and output matrices (please refer to our answer to Reviewer dL7j for a complexity analysis of
> our proposed FC layer and attention module with $G$-invariant metrics).
> However, due to the same reason, Busemann functions associated with $G$-invariant metrics do not fully capture the structure
> of $G$ which might lead to poor performance.
>
> The use of PEM metrics is motivated by the fact that like Log-Euclidean metrics, they turn SPD manifolds into flat spaces
> (i.e., their sectional curvature is null everywhere) which considerably simplifies computation and analysis
> (e.g., Riemannian computations become Euclidean computations in the logarithmic domain [C]).
> PEM metrics are also more general than Log-Euclidean ones.
> On the downside, those metrics do not yield full affine-invariance [C, D]. This might break the geometric stability principle
> that plays a crucial role in geometric deep learning architectures [E].
>
> **References**
>
> [B] S. Helgason. Differential Geometry, Lie Groups, and Symmetric Spaces. ISSN. Elsevier Science, 1979.
>
> [C] Vincent Arsigny, Pierre Fillard, Xavier Pennec, and Nicholas Ayache. Fast and Simple Computations on Tensors with Log-Euclidean Metrics. Technical Report RR-5584, INRIA, 2005.
>
> [D] Lin, Z. Riemannian Geometry of Symmetric Positive Definite Matrices via Cholesky Decomposition. SIAM Journal on Matrix Analysis and Applications, 40(4):1353–1370, 2019.
>
> [E] Michael M. Bronstein, Joan Bruna, Taco Cohen, and Petar Velikovi. Geometric Deep Learning: Grids, Groups, Graphs, Geodesics, and Gauges. CoRR, abs/2104.13478, 2021.

---

> > ### Comment · Reviewer_dvJ8 · 2024-12-01
> >
> > I appreciate the authors' clarifications and recommend incorporating them into the paper. I am comfortable recommending the acceptance.

---

> > > ### Author Response · Authors · 2024-12-01
> > >
> > > We thank the reviewer for the positive feedback and the suggestions for improving the clarity of our paper.

---

### Author Response · Authors · 2024-11-21
**General response**

We thank the reviewers for their insightful comments and suggestions for improving our work. A common weakness pointed out
by the reviewers is the clarity of the implementation of the FC layer and attention module.
Those building blocks will be discussed in detail below.
We propose to add those discussions to the Appendix.

**FC layer**

*Input:* $x \in X$.

*Trainable parameters:* $k_j \in K$, $a_j \in \mathfrak{a},\\| a_j \\|=1$, $p_j \in X,j=1,\ldots,m$, $n \in N$.

*Output:* $y \in X$.

In the case of SPD manifolds, we note that:
- $K = O_m$ (the group of $m \times m$ orthogonal matrices).
- $A$ is the subgroup of $m \times m$ diagonal matrices with positive diagonal entries.
- $N$ is the subgroup of $m \times m$ upper-triangular matrices with diagonal entries $1$.

The dimensions of input, output and trainable parameters then can be inferred accordingly ($a_j \in \mathbb{R}^m,j=1,\ldots,m$).

The computations performed by an FC layer are as follows.

*Step 1:* Compute $\ominus p_j \oplus x = h_j^{-1}gK$ where $p_j=h_jK,x=gK,h_j,g \in G,j=1,\ldots,m$.

*Step 2:* Compute $g_j \in G,j=1,\ldots,m$ such that $k_j^{-1}[\ominus p_j \oplus x] = g_jK$.

*Step 3:* Compute $H(g_j^{-1}),j=1,\ldots,m$ from the Iwasawa decomposition of $g_j^{-1}$, i.e., we need to determine the map
$H: G \rightarrow \mathfrak{a}$ such that $g_j^{-1} = K \exp(H(g_j^{-1})) N$.

*Step 4:* Compute $B_{\xi_j}(\ominus p_j \oplus x),j=1,\ldots,m$ as
\begin{equation*}
B_{\xi_j}(\ominus p_j \oplus x) = \langle a_j, H(g_j^{-1}) \rangle.
\end{equation*}

*Step 5:* Compute the output of the FC layer as
\begin{equation*}
y = n \exp \left ( [-B_{\xi_1}(\ominus p_1 \oplus x), -B_{\xi_2}(\ominus p_2 \oplus x), \ldots, -B_{\xi_m}(\ominus p_m \oplus x) ] \right ) K.
\end{equation*}

**Attention module**

The pipeline of the attention module is given in Fig. 2(a).

*Input:* A sequence of points $x_j \in X,j=1,\ldots,l$.

*Trainable parameters:* Those include trainable parameters of the three FC layers and $c_1,c_2 \in \mathbb{R}$ (used by the attention function).

*Output:* A sequence of points $y_j \in X,j=1,\ldots,l$.

In the case of SPD manifolds, the dimensions of input, output and trainable parameters follow from those in FC layers (see above).

The computations performed by this module are as follows.

*Step 1:* Apply the three FC layers to the sequence of input points $x_j$ and obtain three sequences of points $(q_j)\_{j=1}^l$ (queries),
$(z_j)\_{j=1}^l$ (keys), and $(v_j)\_{j=1}^l$ (values):
\begin{equation*}
(q_j)\_{j=1}^l = f_{lin}^q((x_j)\_{j=1}^l), \hspace{3mm} (z_j)\_{j=1}^l = f_{lin}^z((x_j)\_{j=1}^l), \hspace{3mm} (v_j)\_{j=1}^l = f_{lin}^v((x_j)\_{j=1}^l),
\end{equation*}
where $f_{lin}^q(.)$, $f_{lin}^z(.)$, and $f_{lin}^v(.)$ are the linear transformations performed by the three FC layers.

*Step 2:* Compute the similarities between queries and keys $\bar{\pi}\_{j'j},j',j=1,\ldots,l$ from the Riemannian distance function as
\begin{equation*}
\bar{\pi}\_{j'j} = -c_1d(q_{j'},z_j) - c_2.
\end{equation*}

*Step 3:* Compute the attention weights $\pi\_{j'j},j',j=1,\ldots,l$ as
\begin{equation*}
(\pi_\{j'j})\_{j=1}^l = \operatorname{softmax}\big( (\bar{\pi}\_{j'j})\_{j=1}^l \big).
\end{equation*}

*Step 4:* Perform the midpoint operation to get the sequence of output points $y_{j'},j'=1,\ldots,l$ as
\begin{equation*}
y_{j'} = f_{mid}\big( \\{ v_j,\pi\_{j'j} \\}\_{j=1}^l \big),
\end{equation*}
where $\pi\_{j'j}$ is the attention weight associated with $v_j$.

---

> ### Author Response · Authors · 2024-11-21
> **General response**
>
> # Backpropagation
>
> Below we desribe the backpropagation procedures for FC layers and the attention module in the case of SPD manifolds.
>
> **FC layer**
>
> *Step 1:* Let $p_j = u_js_ju_j^T$ and $x = u_xs_xu_x^T$ be eigen decompositions of $p_j$ and $x$, respectively,
> where $u_j,u_x$ are orthogonal matrices and $s_j,s_x$ are diagonal matrices. Then
> \begin{equation*}
> \ominus p_j \oplus x = s_j^{-\frac{1}{2}} u_j^{-1} u_x s_x^{\frac{1}{2}}K.
> \end{equation*}
>
> Eigen decompositions are differentiable operations in the Tensorflow and Pytorch frameworks.
> Please refer to [A] for a detailed discussion of gradient computations for singular value decompositions and eigen decompositions.
>
> *Step 2:* Compute $g_j = k_j^{-1} s_j^{-\frac{1}{2}} u_j^{-1} u_x s_x^{\frac{1}{2}}$.
>
> *Step 3:*
> Let $g \in G$ and $g = kan$
> with $k \in K$, $a \in A$, and $n \in N$. Then
> \begin{equation*}
> g^Tg = n^Ta^Tk^Tkan = n^Ta^2n = n^Ta(n^Ta)^T.
> \end{equation*}
>
> Let $g^Tg=cc^T$ be the Cholesky decomposition of $g^Tg$ ($c$ is a lower triangular matrix),
> and let $s$ be the diagonal matrix that contains the main diagonal of $c$.
> Then $n = (cs^{-1})^T$ and $a = s$. The map $H: G \rightarrow \mathfrak{a}$ is then given by $H(g) = \log(a)$.
> We see that the map $H$ can be determined from a Cholesky decomposition which is a differentiable
> operation in the Tensorflow and Pytorch frameworks.
>
> *Step 4:* Backpropagation proceeds as normal with the computation of the inner product $\langle a_j, H(g_j^{-1}) \rangle$.
>
> *Step 5:* The output of the FC layer is computed as
> \begin{equation*}
> y = n \exp \left ( 2[ -B_{\xi_1}(\ominus p_1 \oplus x), -B_{\xi_2}(\ominus p_2 \oplus x), \ldots, -B_{\xi_m}(\ominus p_m \oplus x) ] \right ) n^T,
> \end{equation*}
> which involves only a matrix product and so backpropagation proceeds as normal.
>
> **Attention module**
>
> Here we will only describre backpropagation for the two blocks
> "Riemannian distance" and "Midpoint operation". For AttSymSpd-GI, the Riemannian distance used in the ``Riemannian distance" block is given by
> \begin{equation*}
> d(x,y) = \| \log(y^{-\frac{1}{2}}xy^{-\frac{1}{2}}) \|,
> \end{equation*}
> where $x,y \in \operatorname{Sym}_m^+$. The term $y^{-\frac{1}{2}}$ is computed as
> \begin{equation*}
> y^{-\frac{1}{2}} = us^{-\frac{1}{2}}u^T,
> \end{equation*}
> where $y = usu^T$ is an eigen decomposition of $y$, $u$ is an orthonormal matrix and $s$ is a diagonal matrix.
> The $\log(.)$ function is also obtained from an eigen decomposition of its argument.
> As noted previously, eigen decompositions are differentiable operations in the Tensorflow and Pytorch frameworks.
>
> The ``Midpoint operation" block performs the wFM operation under Log-Euclidean framework.
> Let $\{ x_j,w_j \}\_{j=1}^L$ be a set of points $x_j \in \operatorname{Sym}^+_m$ with associated weights
> $w_j \in \mathbb{R}$, where $w_j > 0$ and $\sum\_{j=1}^L w_j = 1$. Then the wFM of these points is given by
> \begin{equation*}
> \operatorname{wFM}(\{ x_j,w_j \}\_{j=1}^L) = \exp\bigg(\sum\_{j=1}^L w_j \log(x_j)\bigg).
> \end{equation*}
>
> The $\exp(.)$ function is computed from an eigen decomposition as $\exp(y) = u\exp(s)u^T$,
> where $u$ is an orthonormal matrix and $s$ is a diagonal matrix. Thus all functions involved in
> the computation of wFM are based on eigen decompositions and so backpropagation proceeds as explained above.
>
> **References**
>
> [A] C. Ionescu, O. Vantzos, and C. Sminchisescu. Training deep networks with structured layers by matrix backpropagation. CoRR, abs/1509.07838, 2015.

---

### Public Comment · ~Bruno_Aristimunha1 · 2025-04-10
**The paper repository is empty.**

Sorry, I couldn't find the code. Is the code in some other link?

---

> ### Public Comment · ~Xuan_Son_Nguyen2 · 2025-04-11
>
> Dear Bruno Aristimunha,
>
> Thank for your interest in our work. We are working on a clean version of the code and will publish it on the provided link soon.
>
> Bests,

---

> > ### Public Comment · ~阮恒睿1 · 2026-04-07
> >
> > Dear Xuan Son Nguyen,
> > I am interested in your SPD MANIFOLDS model. Could you please let me know if the code is available? Thank you for your help.

---

### Meta-Review · Area_Chair_BzAG · 2024-12-17

**Metareview:**

The paper constructs novel neural network layers based on point-to-hyperplane distances in symmetric spaces of noncompact type. All reviews acknowledged the novelty and soundness of the authors' generalization, and are positive in accepting the paper.  In the discussion phase, the authors performed additional experiments on more datasets and clarified the complexity and runtime.

The reviewers also highlighted that the underlying mathematical concepts could be better presented with more intuitions to be accessible to the border audience. In preparing the final version, the authors are recommended to address these comments carefully and implement the proposed revisions in the rebuttal. Please include the computational complexity and runtime analysis, and revise the limitation statement accordingly.

**Additional Comments On Reviewer Discussion:**

All four reviewers participated in the rebuttal phase, and are unanimously positive on the submission.

The reviewers asked for implementation details of the FC/attention layers, which the authors have provided in the rebuttal and agreed to put them in the appendix.

The reviewers asked for computational complexity and runtime comparisons, which the authors have provided in the rebuttal. I have asked the authors to include them in the final version.

---

### Decision · Program_Chairs · 2025-01-22

Accept (Poster)